# Seasonality, molecular epidemiology, and virulence of Respiratory Syncytial Virus (RSV): A perspective into the Brazilian Influenza Surveillance Program

Lucas A. Vianna[1,2]*, Marilda M. Siqueira[3], Lays P. B. Volpini[4], Iuri D. Louro[2], Paola C. Resende[3]

1 Central Laboratory of Public Health of the State of Espirito Santo, Vitoria, Espirito Santo, Brazil, 2 Nucleus of Human and Molecular Genetics/Federal University of Espirito Santo/UFES, Vitoria, Espirito Santo, Brazil, 3 Laboratory of Respiratory Viruses and Measles, WHO NIC, Oswaldo Cruz Institute, FIOCRUZ, Rio de Janeiro, Rio de Janeiro, Brazil, 4 Virology & Infectious Gastroenteritis Laboratory/Federal University of Espirito Santo/ UFES, Vitoria, Espirito Santo, Brazil

* lucasavianna@gmail.com

**Data Availability Statement:** The sequences produced here were deposited on the GenBank platform, under the accession number

## Abstract

### Background

Respiratory Syncytial Virus (RSV) is the main cause of pediatric morbidity and mortality. The complex evolution of RSV creates a need for worldwide surveillance, which may assist in the understanding of multiple viral aspects.

### Objectives

This study aimed to investigate RSV features under the Brazilian Influenza Surveillance Program, evaluating the role of viral load and genetic diversity in disease severity and the influence of climatic factors in viral seasonality.

### Methodology

We have investigated the prevalence of RSV in children up to 3 years of age with severe acute respiratory infection (SARI) in the state of Espirito Santo (ES), Brazil, from 2016 to 2018. RT-qPCR allowed for viral detection and viral load quantification, to evaluate association with clinical features and mapping of local viral seasonality. Gene G sequencing and phylogenetic reconstruction demonstrated local genetic diversity.

### Results

Of 632 evaluated cases, 56% were caused by RSV, with both subtypes A and B co-circulating throughout the years. A discrete inverse association between average temperature and viral circulation was observed. No correlation between viral load and disease severity was observed, but children infected with RSV-A presented a higher clinical severity score (CSS), stayed longer in the hospital, and required intensive care, and ventilatory support more frequently than those infected by RSV-B. Regarding RSV diversity, some local genetic groups

MW026969–MW027004 and MW030961-MW030981, and in the GISAID platform, under the accession number EPI_ISL_549271–EPI_ISL_549327.

**Funding:** This work was funded by Espirito Santo Research and Innovation Support Foundation (FAPES; https://fapes.es.gov.br/), under project Fapes/CNPq n° 05/2017 and by INOVA Fiocruz Program (https://portal.fiocruz.br/programa-inova-fiocruz), under project VPPCB-008-FIO-18. The funders had no role in study design, data collection and analysis, decision to publish, or preparation of the manuscript.

**Competing interests:** The authors have declared that no competing interests exist.

were observed within the main genotypes circulation RSV-A ON1 and RSV-B BA, with strains showing modifications in the G gene amino acid chain.

## Conclusion

Local RSV studies using the Brazilian Influenza Surveillance Program are relevant as they can bring useful information to the global RSV surveillance. Understanding seasonality, virulence, and genetic diversity can aid in the development and suitability of antiviral drugs, vaccines, and assist in the administration of prophylactic strategies.

## Introduction

Respiratory Syncytial Virus (RSV) is the most common pathogen associated with acute respiratory tract infections (ARTI), as well as the main cause of bronchiolitis and pneumonia in infants and small children [1]. RSV infection can cause a range of symptoms, varying from mild upper respiratory tract illness to severe lower respiratory tract infection [2]. The reason for different outcomes is still unclear, however, it can be related to the underlying conditions, genetic or acquired host factors, and/or viral characteristics [3, 4]. Some studies have evaluated the association between viral load and disease severity with significant associations [4, 5]. However, most of these studies did not use standardized methods of viral load measurement, therefore, this relationship must be more carefully evaluated. Understanding the role of the viral load in RSV infection may be a tool to establish its relationship with disease progression, severity, clinical outcome, and drug intervention timeframe [6].

RSV treatment is based only on supportive care and infection prevention is limited to passive immunoprophylaxis (Palivizumab) and case isolation [2]. No approved RSV vaccine is available, but promising candidates are currently in development and in advanced clinical trial phases [7].

RSV strains can be classified into two serogroups: RSV-A and RSV-B [8]. The potential virulence attributed to a specific group remains controversial: some authors have pointed RSV-A [9, 10] or RSV-B [11] as the most virulent subtype, while another study has not found significant differences between them [12]. Multiple genotypes were described for RSV-A and RSV-B, based on the gene G second hypervariable region (HVR-2) [13, 14]. In the past two decades, important genetic changes occurred with the emergence of new RSV-A and RSV-B genotypes: RSV-A ON1 containing a duplication of 72 nucleotides, and RSV-B BA with a duplication of 60 nucleotides in the HVR-2 gene G [14, 15]. These genotypes replaced previous ones and have spread globally. Understanding their genetic diversity may reveal the virus's ability to cause re-infections throughout life, and help in the development of antiviral drugs, diagnostic assays, and vaccines [13].

In 2017, the World Health Organization (WHO) launched the Global Respiratory Syncytial Virus Surveillance Pilot to test the feasibility of using the Global Influenza Surveillance and Response System (GISRS) for RSV surveillance without adversely affecting influenza surveillance [16]. This pilot study results from the global concern about RSV's impact on public health. Brazil, one of four countries in the Americas included in the pilot, has a remarkable respiratory virus surveillance program, however, more data are required for a better understanding of factors such as RSV circulation, evolution, and pathogenicity. In this study, we used the Brazilian Influenza Surveillance Program to analyze the local prevalence of RSV in children with SARI and to evaluate which factors are potentially associated with disease

severity. We also explored the viral seasonality and investigated the influence of climatic factors on circulation. Finally, we conducted a phylogenetic study to understand how the local genetic diversity of RSV behaves when compared to what is observed in the rest of the world.

## Materials and methods

### Population sampling, study period, and location

This study is a retrospective investigation of respiratory samples (nasopharyngeal secretions, tracheal and bronchoalveolar aspirates, and bronchoalveolar lavages) collected from the Brazilian Influenza Surveillance Program over 34 months. (March 7th, 2016, to December 14th, 2018). A total of 632 samples collected from pediatric patients (from 0 to 36 months old) classified as SARI, residents of 60 municipalities in the state of Espirito Santo (ES), were enrolled in this study. ES is located in southeastern Brazil (S1 Fig) and has a territory of 46,074.447 km$^2$, with a population of approximately 4,1 million inhabitants [17]. These samples were screened by real-time RT-qPCR for RSV and Influenza A/B at the ES Central Public Health Laboratory (LACEN/ES), one of 26 Brazilian laboratories that integrate the Brazilian Ministry of Health Influenza Surveillance Program.

### RSV and influenza detection and subtyping

Nucleic acids were extracted from respiratory samples using the PureLink™ Viral RNA/DNA Mini Kit (Invitrogen®, Thermo Fisher Scientific©), according to the manufacturer's protocol. All samples were initially tested for Influenza A and B in a TaqMan® one-step real-time RT-PCR (RT-qPCR) assay using specific primers and probes for influenza (CDC, USA), according to the manufacturer's recommendations. Additionally, an RT-qPCR assay was performed to identify positive RSV cases using a GoTaq® Probe 1-Step RT-qPCR Kit (Promega, Madison, WI, EUA). RSV positive samples (*i.e.* those with cycle threshold [CT] $\leq$ 40) were subtyped using specific primers and probes for RSV-A and RSV-B N genes. In parallel, Ribonuclease P RNA (RNase P) was used as an internal control for each sample and, in all batches, RNA extraction negative control (MOCK) and a PCR negative control (NTC) were used. All primers and probes are described in the S1 Table.

### Clinical and epidemiological data collection

Clinical and epidemiological data were retrieved mainly from the Brazilian Notifiable Diseases Information System (SINAN) database and, in some cases,—when the SINAN form was incomplete—patients' Medical Records were assessed to fill in missing information. The main information retrieved from SINAN were: 1) clinical outcome (recovery or death); 2) hospitalization length of stay; 3) need for oxygen administration and type (invasive or not invasive); 4) intensive care unit (ICU) need and length of stay; 5) clinical characteristics (fever, cough, dyspnea, $O_2$ saturation, respiratory distress, comorbidities), and 6) epidemiological and demographical features (age, town or area of residence).

We have used the Brazilian Ministry of Health's definition of SARI, which is: hospitalized patients with fever and cough or sore throat, and presenting dyspnea or $O_2$ saturation <95%, or respiratory distress [18]. A Clinical Severity Score (CSS) was adapted from Martinello *et al.* [19]. A 6-point scale (0 to 5 spectrum) was used, where 0 indicated the mildest condition and 5 indicated the most severe. ICU admission, hospitalization length of stay $\geq$5 days, oxygen saturation $\leq$95%, and oxygen therapy noninvasive methods accounted for 1 point each. Two points were assigned for mechanical ventilation.

## Viral load quantification

RSV viral load was determined by RT-qPCR using a protocol adapted from Álvarez-Argüelles *et al.* [20], including a synthetic *β-globin* dsDNA as a template. To quantify the RSV copy number, expressed in copies per cell (c/c), we designed a dsDNA containing the annealing regions of RSV primers and probe, as well as the upstream and downstream regions (150 bp). This synthetic DNA was incorporated into a pMA-T plasmid, which was used in the RT-qPCR. Standard curves for absolute quantification of RSV and *β-globin* gene were generated by 10-fold serial dilutions ($10^6$–$10^1$ gene copies), in triplicate. The RSV primers, probe, and thermal cycling protocol adopted were the same used in the diagnostic phase. *β-globin* primers and probe are listed in the S1 Table. All amplification assays were carried out in an ABI 7500 equipment (Applied Biosystems, Foster City, CA, USA). The viral load status was compared with different clinical features and epidemiological data.

## Climate data collection

Climate data (precipitation, temperature, and humidity) of five cities—representing the different geographic regions of the state—were collected daily and kindly provided by the Capixaba Institute of Research, Technical Assistance, and Rural Extension (INCAPER), Vitoria, Espirito Santo, Brazil. The weekly average was accessed by assembling daily data from all collection sites for each epidemiological week (EW). The definition of the RSV epidemic period was based on a previously described protocol [21], which considers RSV outbreak onset, peak, and end. Seasonality onset was defined as the first of 2 consecutive weeks when ≥10% of tested samples for respiratory pathogens were positive for RSV. Similarly, RSV season end was defined when the proportion of positive RSV tests fell below 10% for two consecutive weeks. Peak was determined as the week when the maximum number of RSV positive cases occurred [21].

## Partial amplification and sequencing of glycoprotein gene

RSV-A and RSV-B positive samples were selected for sequencing based on the following criteria: a) cycle threshold (ct) value less than 30, due to the difficulty in sequencing samples with ct higher than this; b) representativeness by collection date; c) distinct clinical outcomes; and d) different viral load values.

The partial gene G amplification (about 730 bp) was performed at LVRS/IOC/FIOCRUZ, the National Influenza Center, by conventional RT-PCR, using the QIAGEN OneStep RT-PCR Kit (Qiagen) and a pair of primers (S1 Table) for each subtype. The reverse transcription was performed at 55°C for 30 minutes and the cDNA was amplified by PCR (40 cycles of 94°C/30 seconds, 60°C /1 minute, 72°C/1 minute and a final extension at 72°C/10 minutes). Amplification was confirmed in a 1% agarose gel. DNA was purified using an ExoSap-IT Kit (Affymetrix, Inc., USA) and submitted for sequencing reaction using a BigDye™ Terminator v3.1 Cycle Sequencing Kit (Applied Biosystems, Foster City, CA, USA) and primers at 3.2 µmolar. The reads were obtained in the ABI 3130XL Genetic Analyzer (Applied Biosystems). Consensus sequences were built from electropherograms comparison with a reference sequence in the software Sequencher 5.1 (Gene Codes Corporation, Ann Arbor, MI, USA). The adopted nomenclature pattern hereon was "hRSV subtype/country/ES-sample number/year."

## RSV genotyping and gene G phylogenetic reconstruction

RSV-A and RSV-B gene G DNA sequences (711 bp and 726 bp, respectively) were used to reconstruct phylogenetic relationships. Genotyping was based on gene G HVR-2, using

RSV-A and RSV-B sequences (336 bp and 318 bp, respectively). Reference sequences of previously described genotypes are shown in the S2 Table. Additionally, to place our sequences in a global context we performed a BLAST search (Basic Local Alignment Search Tool), available at https://blast.ncbi.nlm.nih.gov/Blast.cgi. These sequences (S3 Table) were labeled with country of origin and collection year, and those with more than 99.5% genetic similarity using the CD-HIT tool (http://weizhongli-lab.org/cd-hit/servers.php) were removed from the final dataset. Alignments were conducted using Muscle algorithm, via MEGA 6.0 software [22], and, when necessary, they were adjusted manually. The phylogenetic trees were constructed using the Maximum Likelihood (ML) method, complete deletion for gap or missing data treatment, and 1000 replicates of bootstrap probabilities tools, and analyzed using the Mega 6.0 software. General Time Reversible + Gamma (GTR+G) was the nucleotide substitution model elected for all analyses on JModelTest software, except for RSV-A, where the Tamura-Nei + Gamma (TrN+G) substitution model nucleotide recommended for the analysis was used [23]. Mega 6.0 software was employed to calculate the average pairwise distance (p-distance) and to compare the amino acid changes between Brazilian samples and the reference sequences of ON1 (JN257693) and BA (AY333364).

## Statistical treatment

Statistical analyses were performed using SPSS 20.0 (SPSS, Inc., Chicago, IL) and R v.3.4.4 software. Chi-square, Fisher exact, Mann–Whitney, Kaplan-Meier, and Kruskal Wallis were used whenever appropriate. The Cox regression model was used to assess whether the viral load had a statistically significant effect on length of stay in ICU, and Schoenfeld Residuals were used to check the proportional hazards assumption. To test the association between climate data and RSV circulation, we performed the Spearman correlation test. A p-value of less than 0.05 was considered statistically significant.

## Ethics statement

This project was approved by the Human Research Ethics Committee of the Health Sciences Center of the Federal University of Espirito Santo (UFES), under the number: 018577/2018; CAAE: 84633518.1.0000.5060. The need for consent from parents or guardians was waived by the ethics committee.

## Results

### RSV clinical and epidemiological data

A total of 632 respiratory samples collected from children under 3 years of age were tested by RT-qPCR for Influenza A, Influenza B, and RSV. RSV is the most prevalent pathogen found in these samples (56%; 352/632) (Table 1). From the RSV positive cases, 51% (180/352) were RSV-A, 42% (147/352) were RSV-B, and co-detections with both subtypes were found in 1.4% (5/352). Twenty samples could not be subtyped (5.7%). Influenza frequency was 7.4% (47/632), of which 74% (35/47) were Influenza A H1N1 pdm09, 15% (7/47) were H3N2, and 11% (5/47) were Influenza B. The median age was 4 months old (1–11.0 interquartile range; IQR). Of the positive cases, 99.7% (351/352) were classified as SARI and 14 deaths (4%) were reported.

Table 2 shows patients' clinical features of RSV+ and subtypes. The most frequent symptom reported was cough (93%, 318/341), followed by respiratory distress (88%, 269/307), and fever (86%, 288/336). Seventy-four percent (252/342) of the children needed oxygen therapy and 38% (95/252) of these required mechanical ventilation. The median hospitalization length of

**Table 1. Number of tested samples, RSV positivity, subtype prevalence, and demographic data from each year and the whole study period.**

|  | 2016 n (%) | 2017 n (%) | 2018 n (%) | 2016–18 n (%) |
|---|---|---|---|---|
| **General data** | | | | |
| Sample n˚ | 251/632 (40%) | 135/632 (21%) | 246/632 (39%) | 632/632 (100%) |
| RSV + | 155/251 (62%) | 80/135 (59%) | 117/246 (48%) | 352/632 (56%) |
| RSV - | 96/251 (38%) | 55/135 (41%) | 129/246 (52%) | 280/632 (44%) |
| Flu + | 27/251 (11%) | 6/135 (4%) | 14/246 (6%) | 47/632 (7%) |
| RSV+ deaths | 6/155 (4%) | 5/80 (6%) | 3/117 (3%) | 14/352 (4%) |
| Subtyped samples | 141/155 (91%) | 78/80 (98%) | 113/117 (97%) | 332/352 (94%) |
| **Subtypes** | | | | |
| RSV-A | 58/141 (41%) | 14/78 (18%) | 108/113 (96%) | 180/352 (51%) |
| RSV-B | 80/141 (57%) | 63/78 (81%) | 4/113 (4%) | 147/352 (42%) |
| RSV-A and RSV-B | 3/141 (2%) | 1/78 (1%) | 1/113 (1%) | 5/352 (1.4%) |
| **Demographic data (RSV+)** | | | | |
| Median age (months) | 4 (1–12.0) | 4 (1–10.5) | 3 (1–8.0) | 4 (1–11.0) |
| **Gender** | | | | |
| Male | 72/155 (46%) | 49/80 (61%) | 61/117 (52%) | 182/352 (52%) |

stay was 8 (6–14 IQR) days. Intensive care was needed for 61% (202/333) of patients and the median number of days in ICU was 6 (3–10 IQR). The Kaplan-Meier test was used as a survival analysis technique and revealed that patients' recovery took, on average, 8 days from the date of admission to the ICU (S4 Fig and S5 Table).

**Table 2. Summary of clinical and epidemiological data by RSV+ and each subtype.**

|  | RSV+ n (%) | RSV-A n (%) | RSV-B n (%) | *p-value* |
|---|---|---|---|---|
| **Demographic profile** | | | | |
| Sample number | 352 | 180 | 147 | |
| **Age** | | | | |
| Median age: months (IQR[1]) | 4 (1–11) | 4 (1–10.0) | 4 (1–12.5) | 0.78 |
| **Gender** | | | | |
| Male (%) | 182/352 (52%) | 92/180 (51%) | 78/147 (53%) | 0.725 |
| **Clinical profile** | | | | |
| Fever | 288/336 (86%) | 147/174 (84%) | 124/139 (89%) | 0.223 |
| Cough | 318/341 (93%) | 162/176 (92%) | 134/142 (94%) | 0.418 |
| Dyspnea | 251/331 (76%) | 135/172 (78%) | 97/136 (71%) | 0.148 |
| O$_2$ saturation ≤95% | 169/277 (61%) | 101/150 (67%) | 56/109 (51%) | **0.009** |
| Respiratory distress | 269/307 (88%) | 154/167 (92%) | 96/120 (80%) | **0.002** |
| O$_2$ Therapy | 252/342 (74%) | 138/177 (78%) | 98/143 (68%) | 0.092 |
| *Invasive* | 95/252 (38%) | 56/138 (41%) | 33/98 (34%) | |
| *Noninvasive* | 157/252 (62%) | 82/138 (59%) | 65/98 (66%) | |
| Intensive Care | 202/333 (61%) | 113/168 (67%) | 78/142 (55%) | **0.03** |
| Median hospitalization days | 8 (6–14) | 9 (6–15) | 8 (5–14.0) | 0.15 |
| Median days in Intensive Care | 6 (3–10) | 7 (4–11.0) | 6 (3–9) | 0.13 |
| Deaths RSV+ | 14/352 (4%) | 3/180 (2%) | 8/147 (5%) | 0.07 |

[1]IQR: interquartile range.

Statistically significant values are highlighted in bold. Although the study included 352 patients with RSV, it is possible to observe that the denominators in the clinical profile differ from this number. This occurred because not all clinical data were recorded for all children.

**Table 3. Clinical Severity Score (CSS): Scores varied from 0 to 5.**

| Clinical Severity Score (CSS) | | | | | | |
|---|---|---|---|---|---|---|
| CSS | RSV-A n (%) | RSV-B n (%) | *p-value* | Viral load median (IQR) | n | *p-value* |
| 0 | 1 (1%) | 10 (15%) | **0.003** | 54.06 (6.12–603.61) | 8 | 0.089 |
| 1 | 8 (8%) | 8 (12%) | | 217.41 (96.38–370.56) | 9 | |
| 2 | 19 (20%) | 11 (17%) | | 41.18 (6.53–112.59) | 16 | |
| 3 | 14 (15%) | 15 (23%) | | 17.31 (6.33–125.40) | 14 | |
| 4 | 26 (27%) | 9 (14%) | | 12.05 (4.32–36.63) | 9 | |
| 5 | 28 (29%) | 13 (20%) | | 11.81 (1.14–54.24) | 18 | |

Higher values indicated more severe illness. Need for ICU, $O_2 \leq 95\%$, hospitalization length of stay >5 days, and requirement of $O_2$ therapy accounted for 1 point each. The need for mechanical ventilation accounted for 2 points. Patients infected with RSV-A were most commonly classified into the most severe scores.

When compared to RSV-B, patients affected by RSV-A showed a higher frequency of respiratory distress (92% vs 80%, p = 0.002), and more often manifested $O_2$ saturation $\leq 95\%$ (67% vs 51%, p = 0.009) and higher requirement for intensive care (67% vs 55%, p = 0.03). Our data also indicate that patients affected by RSV-A stayed one day longer in the hospital and intensive care units than those affected by RSV-B, however, these data were not statistically significant. Lastly, the RSV-A viral load showed more than twice the number of virus copies per cell (median = 57.41 copies/cell) than RSV-B (median = 27.35 copies/cell). RSV-A CSS median was 4 and RSV-B's was 3, and children infected by RSV-A were most frequently classified in higher severity scores than those infected by RSV-B (Table 3).

## Viral load

A total of 156 (44%) samples were submitted to the viral load analysis (Table 4). According to age, the median viral load was higher in children with 4 to 6 months of age (63.0 cop/cell, p = 0.007). Regarding patients' clinical conditions, we found a lower viral load in patients with fever (26.15 cop/cell) than in those without (111.29 cop/cell; p<0.001), and a higher viral load (70.24 cop/cell) in patients without the need for oxygen therapy (22.69 cop/cell; p = 0.02). Deceased patients had a lower viral load (2.80 cop/cell; p = 0.02) in comparison to the others (37.96 cop/cell). Although lacking statistical support (p = 0.089), a noteworthy observation is the tendency towards a lower viral load in patients with elevated CSS. The viral load analysis was performed regardless of the time between symptoms onset and date of collection, which, in theory, could cause an analytical bias due to the natural decrease in viral load over the course of the disease. However, a segmented analysis (0–3; 4–7 and >7 days between symptom onset and sample collection) revealed very similar results. Furthermore, of the 156 samples used to measure viral titers, only 26 (16%) were collected 7 days after symptoms onset. Therefore, we opted to maintain full sampling for viral load analysis.

The Cox regression model showed that the viral load did not have a statistically significant effect on ICU length of stay (p = 0.29; 95% CI: 0.99–1.00). Schoenfeld Residuals (S5 Fig and S6 Table) showed that the proportional hazards assumption was met (p = 0.95).

## Viral seasonality and climatic analysis

In 2016 and 2017, the RSV season started in the 12[th] EW (March, early fall season), peaked between the 16[th]–20[th] EW, and ended in the winter season, between the 31[st]–32[nd] EW (Fig 1; S7 Table). In 2018, the beginning of RSV seasonality was observed earlier, with the first cases occurring in the 3[rd] EW, (January, in the middle of summer). The peak took place in the 14[th]

**Table 4. Comparison of viral load values between gender, age, outcome, and clinical condition.**

| Demographic data | | | | |
|---|---|---|---|---|
| **Parameter** | | **N** | **Median (IQR[1])** | **p-value** |
| **Gender** | Male | 78 | 51.40 (8.13–265.31) | 0.08 |
| | Female | 78 | 24.63 (4.46–88.29) | |
| **Age (months)** | 0–3 | 86 | 51.40 (6.12–152.90) | **0.007** |
| | 4–6 | 22 | 63.09 (32.12–211.67) | |
| | 7–12 | 21 | 39.29 (2.32–236.91) | |
| | >12 | 26 | 7.77 (1.72–36.92) | |
| **Outcome** | Recovery | 130 | 37.96 (6.72–122.71) | **0.02** |
| | Death | 7 | 2.80 (0.04–21.49) | |
| **Clinical data** | | | | |
| **Fever** | Yes | 121 | 26.15 (4.33–104.46) | **<0.001** |
| | No | 27 | 111.29 (51.80–408.21) | |
| **Cough** | Yes | 144 | 41.53 (4.86–148.15) | 0.59 |
| | No | 7 | 11.52 (7.98–106.29) | |
| **Dyspnea** | Yes | 106 | 37.96 (3.91–154.88) | 0.69 |
| | No | 40 | 42.05 (8.58–120.16) | |
| **O$_2$ saturation ≤ 95%** | Yes | 71 | 26.41 (3.95–150.65) | 0.40 |
| | No | 51 | 50.16 (8.36–196.81) | |
| **Respiratory distress** | Yes | 115 | 39.29 (4.78–150.13) | 0.27 |
| | No | 18 | 75.69 (12.66–214.26) | |
| **Days of hospitalization** | 1–4 | 20 | 79.36 (11.10–245.08) | 0.20 |
| | 5–8 | 49 | 39.45 (11.89–176.21) | |
| | >8 | 54 | 24.42 (4.08–78.04) | |
| **Ventilatory support** | *No* | 48 | 70.24 (11.41–342.96) | **0.02** |
| | *Yes (total)* | | 22.69 | |
| | *Yes—noninvasive* | 65 | 26.41 (6.26–105.11) | 0.35 |
| | *Yes—invasive* | 40 | 17.31 (3.95–68.70) | |
| **Intensive Care** | Yes | 82 | 30.01 (4.41–113.44) | 0.73 |
| | No | 67 | 39.29 (6.90–154.61) | |
| **Days of Intensive Care** | 1–4 | 20 | 34.74 (3.60–226.28) | 0.547 |
| | 5–8 | 16 | 16.27 (2.09–106.22) | |
| | >8 | 24 | 36.24 (9.10–106.65) | |
| **Days of symptom until collect** | 0–3 | 51 | 36.63 (5.99–220.48) | 0.19 |
| | 4–6 | 67 | 39.98 (7.65–135.24) | |
| | 7–9 | 24 | 19.98 (0.53–77.39) | |
| | >9 | 12 | 10.45 (3.92–50.98) | |
| **Subtype** | RSV-A | 64 | 57.41 | **0.03** |
| | RSV-B | 76 | 27.35 | |

[1] IQR: interquartile range.

Statistically significant p-values are highlighted in bold.

EW and the end occurred in the 27[th] EW. Thus, the RSV seasonal period in 2016, 2017, and 2018 lasted 20, 19, and 24 weeks, respectively.

Precipitation rate and relative humidity percentage have not been shown to influence the distribution of RSV cases by Spearman's correlation test (p = 0.55 and 0.11, respectively). The

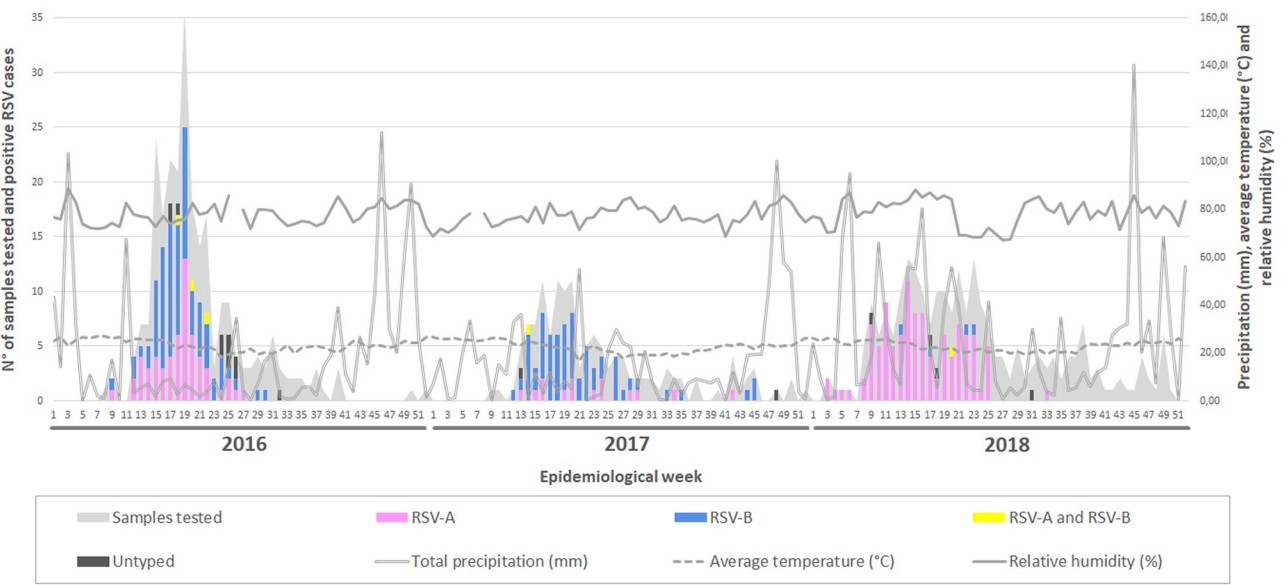

**Fig 1. Circulation of RSV-A and RSV-B between 2016 and 2018 in Espirito Santo State.** The X-axis shows the epidemiological weeks (EW) for each year. The primary Y-axis displays the number of positive cases for each of the subtypes and the secondary Y-axis shows the values of the climatic variables. The gray zone indicates the total number of samples tested in each EW.

mean temperature, however, showed a minor and inverse correlation with RSV infections (-0.16; p = 0.05).

Although RSV-A and RSV-B co-circulated each year, it is noteworthy that the subtype distribution changed over the years. In 2016, RSV-B predominated (n = 80; 58%) over RSV-A (n = 58; 42%). In 2017 this difference increased, and RSV-B was responsible for 82% of the cases (n = 63). Finally, in 2018, there was a shift in this pattern and almost all RSV cases were caused by RSV-A (n = 108; 96%).

## Phylogeny of RSV and genetic analysis

The phylogenetic reconstructions revealed that 36 RSV-A were classified as GA2.ON1 genotype and 21 RSV-B were classified as BA genotypes, based on the 2nd HVR (S2 and S3 Figs). Some local genetic groups of both genotypes and a slightly higher diversity among the RSV-A strains (p-distance = 1.8%) were observed in comparison to RSV-B (p-distance = 1.6%) (Figs 2 and 3).

RSV-A ES Brazilian strains, from 2016 to 2018, are clustered with strains that circulated in North America, South America, Asia, Africa, and Oceania, from 2011 to 2018. A Brazilian main local cluster BR.1 (L142S, L274P, Y304H, and T320A) circulated in ES state, from 2016 to 2018. Additionally, two new subclusters, BR.1.1 (E106G,) and BR.1.2 (N103T, S144I, E224V, S270P, and/or P298L) were detected co-circulating in the ES state in 2018. Amino acid substitutions, compared with the RSV-A GA2.ON1 reference strain (JN257693), can be observed in the S8 Table. The average CSS inside the BR.1 cluster was 2.84, while the average in the rest of the BR strains was 3.78, showing that the BR.1 cluster may be more associated with lower severity disease than the other strains. The viral load seemed to be higher on BR.1 strains when compared to other Brazilian strains.

RSV-B gene G phylogenetic reconstruction (Fig 3) revealed that Brazilian strains from 2016 to 2018 belonged to a cluster containing global strains circulating since 1999. ES Brazilian strains were distributed through this main cluster and presented punctual amino acid

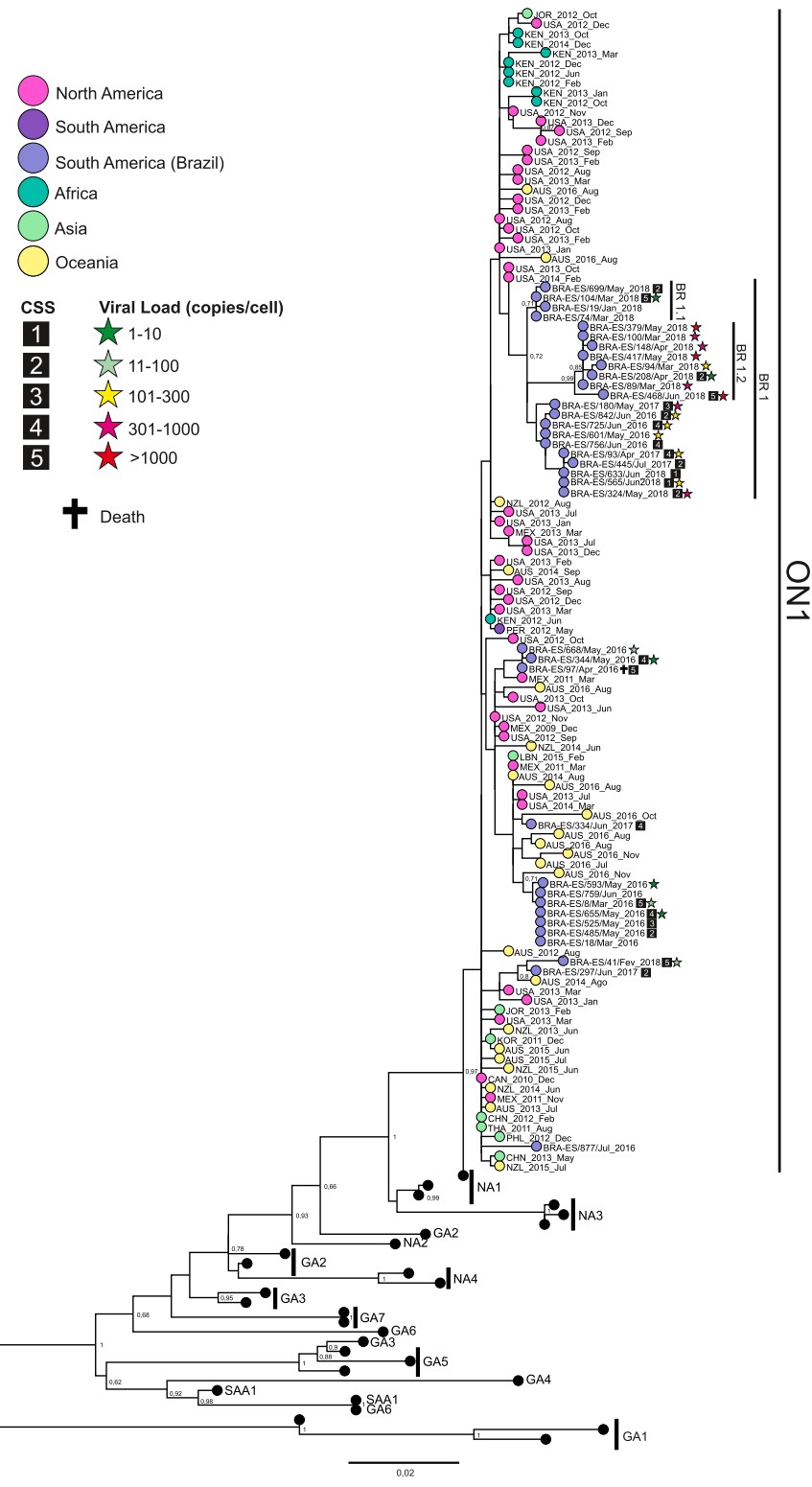

**Fig 2. RSV-A phylogenetic tree.** The tree was built using the maximum likelihood method on MEGA 6.0 software from a MUSCLE alignment of G gene sequences of 711 bp. Previously published sequences from known genotypes were retrieved from the NCBI database. Numbers from 1 to 5 within the squares indicate the patients' CSS. The cross indicates patients who died due to RSV infection. The stars indicate the viral load, categorized by color (in copies per cell).

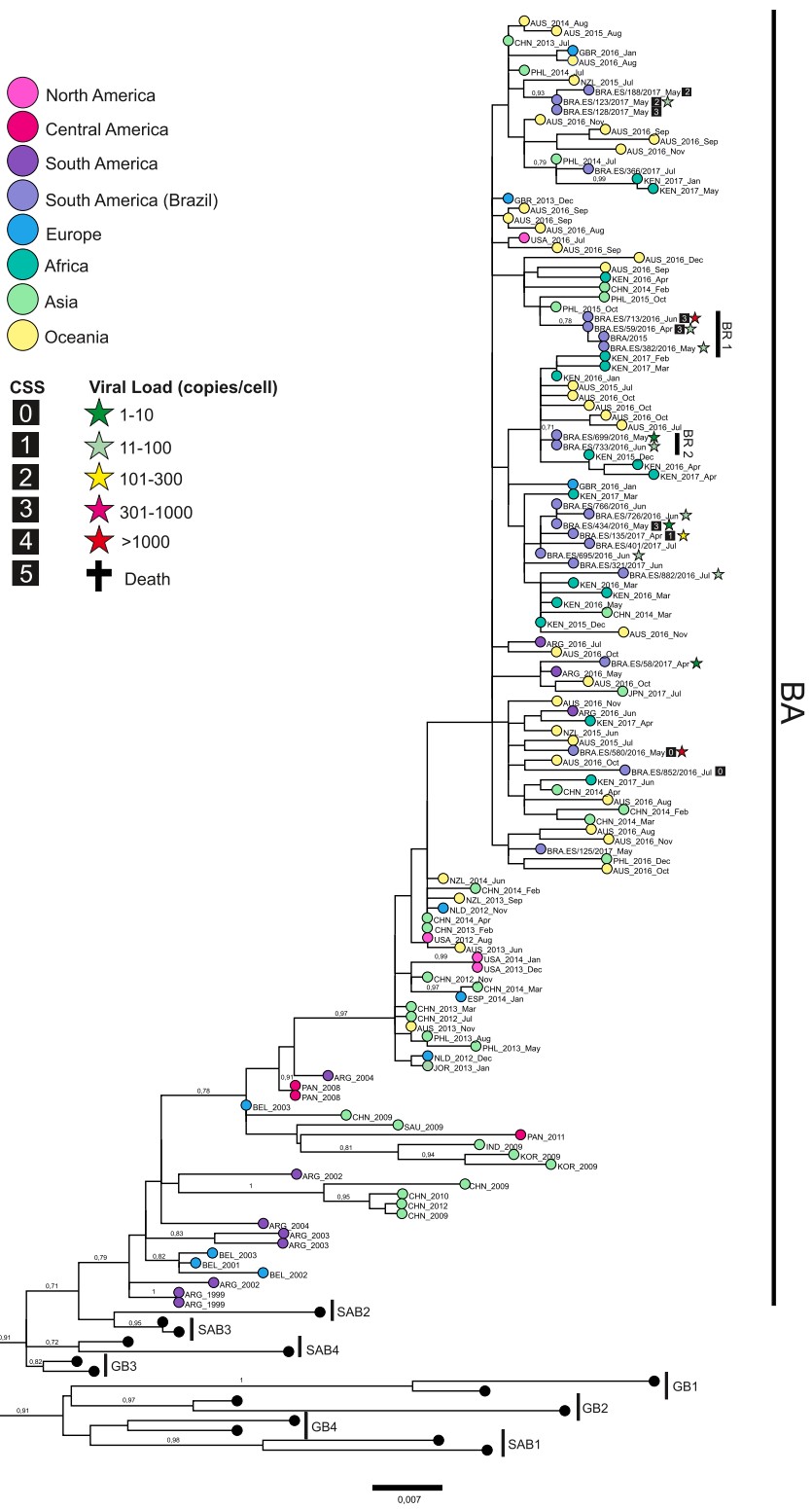

**Fig 3. RSV-B phylogenetic tree.** The tree was built using the maximum likelihood method on MEGA 6.0 software from a MUSCLE alignment of G gene sequences of 726 bp. Previously published sequences from known genotypes were retrieved from the NCBI database. Numbers from 1 to 5 within the squares indicate the patients' CSS. The cross indicates patients who died due to RSV infection. The stars indicate the viral load, categorized by color (in copies per cell).

substitutions, some of them with a potential loss of O-glycosylation, such as T229N and/or S287F (strains from 2017). Inside the main cluster, some local subclusters were observed, such as the BR.1 (S101G loss glycosylation site, P217L, and T248A loss glycosylation site) and BR.2 (G136S and S269P), in samples from 2016, revealing a large diversity among RSV-B viruses circulating in the ES State during that year. Additionally, two strains from 2017 presented an insertion of tree nucleotides at codon 228. All these amino acid substitutions, compared with the RSV-B BA reference strain (AY333364), are described in the S9 Table. CSS and viral load data were unavailable for most of the RSV-B sequences, therefore, we could not compare those data with the genetic strains observed.

## Discussion

In this paper, we investigated RSV features using the Brazilian Influenza Surveillance Program and addressed some RSV issues listed in the WHO global RSV surveillance pilot objectives [16], such as the RSV burden in hospitalized children and mapping of local seasonality. Additionally, we described the molecular characteristics of gene G which revealed RSV-A and RSV-B local clusters co-circulating in Brazil.

### RSV is prevalent in Brazilian children with SARI

RSV prevalence in different Brazilian regions is highly diverse, ranging from 7.7% to 77.6% [24–26]. In the ES state, from 2016 to 2018, the prevalence in hospitalized children up to 3 years of age was 56%. These differences are probably related to the use of diverse methods of RSV detection (*e.g.* RT-PCR or immunofluorescence) or patient inclusion criteria (*e.g.* age, symptoms, period of the year). During the 1997–98 season, Checon *et al.* found a prevalence of 28% in the capital of ES State [26]. This lower prevalence in comparison to our study can be attributed to the less sensitive method used by the authors (immunofluorescence) and a broader target population age (children ≤ 5 years old).

In our study, the median age of four months in hospitalized children with RSV confirms the higher prevalence in children younger than one year of age [2], which justifies why RSV vaccine candidates are aiming to protect, primarily, infants and young children [7]. Although the median hospitalization length of stay found here is similar to some other studies [27, 28], notably, most of them report a shorter duration [1, 4, 25]. One hypothesis that could explain this finding is the fact that all children included in our study were diagnosed with SARI, which makes our study group a cohort with severe RSV infection. Another hypothesis is linked to the possibility that most of the children in the study had an infection in the lower respiratory tract. Aerosol transmission increases the chances of inhaling viral particles in the lower airways, while larger droplets are retained in the upper airways [29]. Naturally, aerosol infections tend to trigger a more severe course of infection [30].

### The subtype but not the viral load appears to be associated with disease severity

RSV infection can cause a range of clinical outcomes [2], but factors attributed to a worse outcome remain unclear [3, 4]. Several studies have shown that the male gender is a risk factor for RSV infection [2], while others have not observed such a connection [31]. Although not statistically significant, we observed that male children were slightly more affected than female, which could support the hypothesis that male children are at higher risk. Nevertheless, the CSS median was three for both genders.

Although some authors have found no correlation between subtypes and disease severity [32, 33], many others indicate RSV-A as the most virulent subtype [9, 10, 12, 34, 35]. We have

found that children infected with RSV-A revealed a higher clinical score index (CSS median = 4)–therefore, a more severe disease—when compared to those infected with RSV-B (CSS median = 3). Children infected by RSV-A required $O_2$ therapy more often than those infected by RSV-B and, of all children who needed $O_2$ therapy, those affected by subgroup A needed mechanical ventilation more frequently. Although these data did not have statistical support, other studies found the same connection [9, 10]. Our data also show that children infected by subgroup A required ICU more often (p = 0.03) and remained hospitalized and in ICU a day longer, on average, when compared to those infected by RSV-B, which is in agreement with previous studies [35, 36]. Notwithstanding, we highlight that only one genotype was found for each subtype (ON1 and BA), thus, those differences in severity could be a consequence of differences in the genotype's virulence, rather than in the subtype's.

The correlation between disease severity and viral load remains controversial. While several authors have shown that the severity of the infection follows the viral load [4, 5, 37, 38], others have not [7, 12, 33]. Some studies found an association between viral load and symptom frequency, but not severity itself [39, 40]. Viral load measurement methods are widely variable between studies: some authors use plaque assay [4] or semi-quantitative analyses, such as ct [5, 7, 32], others use quantitative methods [38–41]. Moreover, most studies that use quantitative methods do not normalize the measurements. Respiratory samples are naturally heterogeneous and the collection technique can influence viral genome concentration [38].

In this study, we used a standardized method for measuring the viral load. Interestingly, we found a lower viral load in patients with fever (p = 0.00), with the need for ventilatory support (p = 0.02), and in those who died (p = 0.02). Our data conflict with previous studies that demonstrated a positive association between viral load and the presence of cough, fever [39], and the need for intubation [37]. However, two recent studies reported a higher viral load in less severe RSV disease [42, 43]. Piedra *et al.* observed a positive correlation between viral load and mucosal concentration of proinflammatory cytokines that may suggest that high RSV loads can protect from disease progression due to the promotion of an early robust innate immune response [42, 43]. Conflicting results between studies could be attributed to the different methods used to calculate viral load, various study designs, and indicators of disease severity.

## The seasonal period of RSV may fluctuate and its circulation is slightly associated with temperature

In temperate countries, RSV peak activity occurs in the winter and several studies have shown the connection between cold temperatures and viral circulation [44]. In contrast, in tropical countries, there is a wide range of variability in the timing and duration of epidemics and the correlation between climatic factors and viral activity is controversial [21, 45]. Although in the Southern Hemisphere the RSV wave usually starts between March and June and decreases between August and October [21], in Brazil, a continental country with five geographic regions, a wide variation in the seasonality is seen, such as those observed in the northeastern [46] and southern [47] regions.

Here, we showed that RSV's activity was very similar between the 2016 and 2017 seasons, with the circulation onset occurring in March (EW 12) and ending in July/August (EW 31–32), during the winter season. These data are in accord with the Brazilian Society of Pediatrics, which recommends the administration of Palivizumab from February to July [48]. Nonetheless, in 2018, we observed an early occurrence of the seasonality onset by nine weeks, with the beginning of circulation occurring in January (summer season) and the end taking place in the Fall instead of Winter.

In the southeastern region, it was observed that the RSV peak usually happens in early April [49]. Our data shows that, in 2016, the RSV peak occurred in May, suggesting subtle differences even inside the same geographical region. In 2018, there was an extension of RSV's seasonality duration by 4.5 weeks when compared to the average in 2016–2017. Those observations are especially worrisome since major variations could make a preventive measure harder to implement. Understanding local epidemics is important in managing the time of prophylaxis, supporting vaccine development, and following morbidity and mortality caused by RSV infection [44]. Thus, establishing RSV surveillance in real-time may allow for the identification of patterns and possible variations in prophylaxis time. RSV seasonality usually lasts five to six months [21]. In our study, the longest seasonal period occurred in 2018 (6 months), followed by 2016 (5 months) and 2017 (4.75 months). Interestingly, the prevalence of RSV-A was high in 2018 (96%), medium in 2016 (41%), and low in 2017 (18%). These data reinforce the theory that RSV-A may lengthen the seasonality [50].

Climatic factors, such as humidity, rainfall, and temperature have been assumed to impact RSV seasonality [44, 51]. However, this association remains controversial. An inverted correlation between RSV circulation, temperature, and humidity was observed in a Brazilian study, carried out in the state of São Paulo [52]. In this study, a minor correlation was found between temperature decrease and case number increase. However, no correlation was found concerning humidity or precipitation.

## ON1 and BA were the only genotypes detected

All RSV-A isolates were ON1 genotype and all RSV-B were BA, which confirms the fast-global dissemination of RSV with nucleotide duplication. These findings are consistent with recently published reports performed in other countries, such as the Philippines [53], Kenya [54], Italy [55], USA, and Puerto Rico [56].

Overall p-distance during the study period in RSV-A was 1.8%. A recent study observed an overall p-distance of 1.4% within ON1 [13]. A noteworthy observation is the fact that in 2017 we found the lowest prevalence of RSV-A in ES (18%), and yet, still, the highest genetic diversity. Phylogeny showed that 2017 strains were distributed in almost all genetic clusters, which showed high diversity that year. RSV-A phylogenetic analysis revealed ongoing genetic changes, with BR.1 grouping the most recent strains, suggesting that BR.1 strains may be under positive selective pressure. Changes in the circulation of RSV strains have been considered a mechanism for evading immune response generated by previous strains, which possibly allows for re-infections to occur [57].

As demonstrated, in 2018 RSV-B was responsible for only 4% of cases. Therefore, the phylogenetic analysis did not include any RSV-B samples from that year. Older strains, from 2009 to 2014, are positioned at the base of the BA cluster, however, sample strains collected between 2015 and 2018 did not form genetic groups related to the year of collection. This observation may suggest an absence of positive pressure.

Although we found clusters composed exclusively of ES samples, it is necessary to expand the sequencing of RSV samples globally to verify if there is, in fact, the formation of local genetic groups or if the observation is caused by a sample bias.

Previous studies showed that a large part of the genetic variability between RSV strains comes from changes in the O-glycosylation profile and that this may be associated with an evolutionary mechanism of immune response evasion [58]. Here, we investigated and listed strain amino acid substitutions and also those shared within and between clusters. However, we did not carry out an in-depth analysis to understand the role of these mutations, as our objective was purely observational. Among the mutations found, one of the most interesting was the

insertion of three nucleotides at codon 228 in RSV-B. Further studies are essential to understand virus evolution and pathogenicity mutation consequences.

Limitations of this study include the fact that the majority of patients had an acute infection, thus, the prevalence found refers only to SARI, and the absence of a mild infection group prevents further analysis of severity influencing factors. Furthermore, clinical data were taken from notification forms, which often contain inconsistencies and missing data. Despite those caveats, we believe the data provide valuable epidemiological, genetic, and clinical information on RSV.

## Conclusion

In this study, we observed a high prevalence of RSV in children under three years of age even when using the Brazilian Influenza Surveillance Program. This result is important because it shows that the establishment of global RSV surveillance within the Influenza surveillance system allows for the detection of a large number of cases. Our data suggest that RSV-A is, in fact, more virulent than RSV-B. Notably, no correlation between viral load and disease severity was observed. The observation of a marked early onset of the seasonal period is worrisome since this can make it difficult to administer prophylactic measures at the right time, however, it is necessary to expand the historical series of seasonality in the state of Espirito Santo. The average temperature was the only climatic factor to show interference with the viral circulation. Our data show the annual co-circulation of RSV-A and RSV-B, however, with considerable fluctuations in the prevalence of subtypes. ON1 and BA were the only genotypes found in the studied period, which corroborates a series of recent studies. The establishment of a global and standardized real-time RSV surveillance may allow for the collection of data that will help to understand the complex mechanisms of viral evolution and will facilitate the development of future vaccines and antiviral drugs.

## Supporting information

**S1 Fig. Map of the Espirito Santo State (Brazil) and its federal highways.** The state is divided into 78 municipalities, of which 60 were represented by children with SARI and 46 with children with confirmed RSV infection. The colors of the municipalities represent the number of positive RSV cases.
(TIF)

**S2 Fig. RSV-A phylogenetic tree based on 336 bp of the HVR-2 of G gene.** The tree was built using the maximum likelihood method on MEGA 6.0 software from a MUSCLE alignment, with some manual editions. Reference sequences from each described genotype were downloaded from the NCBI GenBank and used in the phylogenetic reconstruction. The genotypes were classified by colors and all ES strains were grouped within the ON1 genotype.
(TIF)

**S3 Fig. RSV-A phylogenetic tree based on 318 bp of the HVR-2 of G gene.** The tree was built using the maximum likelihood method on MEGA 6.0 software from a MUSCLE alignment, with some manual editions. Reference sequences from each described genotype were downloaded from the NCBI GenBank and used in the phylogenetic reconstruction. The genotypes were classified by colors and all ES strains were grouped within the BA genotype.
(TIF)

**S4 Fig. Survival curve in relation to ICU length of stay estimated by the Kaplan-Meier test.** Given the small number of deaths, it was necessary to modify the analysis to assess the

likelihood of cure.
(TIF)

**S5 Fig. Graph of Schoenfeld residues: There was no marked trend, indicating that the premises for the application of the Cox model were met.**
(TIF)

**S1 Table. Primers, probes, and DNA fragments used in the study.** "F", "R", and "P", represent the sequence of the forward and reverse primers, and the probe, respectively. A synthetic DNA fragment from RSV was included in a pMA-t vector.
(DOCX)

**S2 Table. List of the sequences used to build the phylogeny based on HVR-2 of gene G for both subtypes RSA-A and RSV-B.**
(DOCX)

**S3 Table. List of the sequences used to build the phylogeny based on gene G for both subtypes RSV-A and RSV-B.** The collection date of some sequences was unavailable.
(DOCX)

**S4 Table. General table that provides all epidemiological, clinical, and climatic data of the study.**
(XLSX)

**S5 Table. Cox (proportional hazards) regression: Given that the p-value is >0.05, it can be inferred that the viral load has no significant effect on ICU length of stay.**
(DOCX)

**S6 Table. Proportional hazards assumption test: The premises for the application of the Cox model were met.**
(DOCX)

**S7 Table. Duration and climatic characteristics of RSV seasonality in the years studied.**
(DOCX)

**S8 Table. List of amino acid changes in RSV-A.** Residues in blue and red show potential losses and gains of O-glycosylation sites, respectively.
(XLSX)

**S9 Table. List of amino acid changes in RSV-B.** Residues in blue and red show potential losses and gains of O-glycosylation sites, respectively.
(XLSX)

## Acknowledgments

We would like to thank Liliana Cruz Spano for her significant theoretical and experimental support to this work, Beatriz Alves Vianna, Fabíola Karla Correa Ribeiro, and Mirella Martins Tostes for their assistance in editing and improving the language, all researchers who upload genetic sequences in the public genetic database—GenBank, patients, parents and guardians, the Espirito Santo State Health Department, and the Brazilian Ministry of Health, represented by the Influenza Technical Group.

## Author Contributions

**Conceptualization:** Lucas A. Vianna, Marilda M. Siqueira, Lays P. B. Volpini, Iuri D. Louro, Paola C. Resende.

**Data curation:** Lucas A. Vianna, Paola C. Resende.

**Formal analysis:** Lucas A. Vianna, Lays P. B. Volpini, Iuri D. Louro, Paola C. Resende.

**Funding acquisition:** Marilda M. Siqueira, Iuri D. Louro, Paola C. Resende.

**Investigation:** Lucas A. Vianna.

**Methodology:** Lucas A. Vianna, Marilda M. Siqueira, Paola C. Resende.

**Project administration:** Lucas A. Vianna, Iuri D. Louro, Paola C. Resende.

**Resources:** Marilda M. Siqueira, Lays P. B. Volpini, Iuri D. Louro, Paola C. Resende.

**Supervision:** Marilda M. Siqueira, Iuri D. Louro, Paola C. Resende.

**Validation:** Lucas A. Vianna.

**Writing – original draft:** Lucas A. Vianna, Paola C. Resende.

**Writing – review & editing:** Lucas A. Vianna, Marilda M. Siqueira, Lays P. B. Volpini, Iuri D. Louro, Paola C. Resende.

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
