## [Decision Letter · Decision Letter 0]

16 Dec 2020

PONE-D-20-30854

Landscape of Respiratory Syncytial Virus (RSV): a perspective into the Brazilian Influenza Surveillance Program

PLOS ONE

Dear Dr. Vianna,

Thank you for submitting your manuscript to PLOS ONE. After careful consideration, we feel that it has merit but does not fully meet PLOS ONE’s publication criteria as it currently stands. Therefore, we invite you to submit a revised version of the manuscript that addresses the points raised during the review process.

I have received the reviews of your manuscript. While your paper addresses an interesting question, the reviewers stated several concerns about your study and did not recommend publication in its present form. Both reviewers voice a number of concerns regarding the presentation as well as data analysis, and these comments need to be addressed carefully. Please see reviewers’ insightful comments below.

The quality of the language needs to be improved, there are quite a few awkward sentences, typo and grammatical errors throughout the manuscript. Please have a fluent, preferably native, English-language speaker thoroughly copyedit your manuscript for language usage, spelling, and grammar.  

We look forward to receiving your revised manuscript.

Kind regards,

Baochuan Lin, Ph.D.

Academic Editor

PLOS ONE

Journal Requirements:

2. We note that you included minors (age<18) in your study.

Please provide additional details regarding minors consent.

In the ethics statement in the Methods and online submission information, please ensure that you have specified whether you obtained consent from parents or guardians. If the need for consent was waived by the ethics committee, please include this information.

Reviewers' comments:

Reviewer's Responses to Questions

**Comments to the Author**

1. Is the manuscript technically sound, and do the data support the conclusions?

Reviewer #1: Partly

Reviewer #2: Partly

2. Has the statistical analysis been performed appropriately and rigorously? 

Reviewer #1: Yes

Reviewer #2: Yes

3. Have the authors made all data underlying the findings in their manuscript fully available?

Reviewer #1: Yes

Reviewer #2: Yes

4. Is the manuscript presented in an intelligible fashion and written in standard English?

Reviewer #1: No

Reviewer #2: Yes

5. Review Comments to the Author

Reviewer #1: This manuscript has scientifically valid information that is worth publishing. The authors have done a lot of work and produced data. However, the manuscript cannot be accepted at its present form.

Title - Revise the title to reflect the key findings of the research.

Introduction - Revise the introduction and make it shorter and its current form it distracts the reader.

Methodology - Organize the Methodology - Study population, experimental methods - RT-qPCR, sequencing etc. in a brief form that gives a good understanding of the sequence of events with relevant methodology.

Results - They need to be organized in the order of appearance as in the Methodology. You do not have to replicate all the information given in the Tables in the texts. Redundancy also distracts the reader and I found it difficult to organize the results to understand the authors' way of cohesion.

Discussion - Again follow the order of your results in Discussion.

Overall - You must revise this manuscript shortening certain sections and organizing the manuscript from I to D. Otherwise the results produced cannot be understood by the authors.

Language must be clear, correct, and unambiguous. At its current form it is difficult to follow the authors. Please also look into typographical or grammatical errors when your revise the manuscript. You may ask a native speaker to read the manuscript after fixing all the issues indicated.

Please follow the PLOS ONE formatting guidelines well before you submit after revision.

Reviewer #2: Reviewer #1

Summary

The manuscript by Vianna et al. describes an evaluation of the role of viral load and genetic diversity of RSV on disease severity in hospitalized children under 3 years old identified using the Brazilian Influenza Surveillance Program and the influence of climate factors on RSV seasonality in Espirito Santo State of Brazil. The authors present data showing no correlation between viral load and disease severity and that some clinical features of disease severity were significantly higher among patients with RSV-A compared to RSV-B. The manuscript is very extensive and the authors often reference Supplemental information to support interpretations and main messages (e.g. clinical severity scores). Genetic diversity analyses revealed local co-circulating clusters of RSV-A (all ON1) and RSV-B (all BA) during the period of study (2016-2018), with severity of disease impact investigated for RSV-A strains only due to lack of CSS data for RSV-B strains. Finally, climate factors including precipitation rate, percent humidity, and mean temperature showed no significant correlation on RSV seasonality between 2016-2018. The authors recognize the importance of global RSV surveillance and highlight the potential of the Global Influenza Surveillance and Response System (GISRS) as a platform to collect data that could follow temporal evolution of RSV and potentially support the development RSV vaccine and antivirals. Given the limited availability of RSV surveillance data in Brazil, this study provides recent information to better understanding of seasonality and the impact of RSV molecular epidemiology on disease severity relative to other areas of the world

Major Comments for the Author

1. The authors present generalized conclusions that are not specific to the Brazilian aspect or timeframe on which this study is based. Key results to support the identified objectives are not highlighted in the abstract or the conclusion (e.g. the influence of climate factors on RSV seasonality and the role of genetic diversity of RSV on disease severity). Clinical severity scores referenced in the abstract and results to support interpretations of the role of viral load and genetic diversity of RSV on disease severity, should be presented in the main tables/figures of the manuscript as opposed to supplemental. The authors should revise these areas and sharpen the focus of their Discussion through reduction to improve readability and presentation of key messages of RSV surveillance in Brazil between 2016-2018 relative to previous observations in Brazil or other parts of the world during the similar timeframe.

2. The current title (and abstract) fail to address the presented timeframe of RSV surveillance or what aspects of “landscape” or “perspectives” the authors are referring to relative to their objectives and results. The authors should consider revision.

3. Figure 2 and Figure 3 are out of focus and uninterruptable for review. The authors should revise.

4. Line 90-92, 130-133, and 270-276. The authors statement of “seasonal oscillation” (Line 90-92) is not supported by their main observations (Line 270-276) from Fig. 1, and in turn, their statement regarding “creating difficulties for determination of the most appropriate period to start prophylaxis” is not substantiated and is in contrast to their later statement of “recommends the administration of palivizumab from February to July” (Line 359). The authors state (Line 130-133) that “seasonality onset and end were defined as the first and last of 2 consecutive weeks, respectively, when the number of RSV cases exceeded 10% of the number detected during the RSV peak week” and reference Obando-Pacheco et al 2018 [21]. However, Obando-Pacheco et al 2018 states that “the onset of RSV season was defined as the first 2 consecutive weeks when >10% of the total tested samples for respiratory pathogens were positive for RSV. The end of the RSV season was defined similarly as when the proportion of positive RSV tests fell below 10% for 2 consecutive weeks.”. Given the impact of molecular testing on determining RSV seasonality, the authors should revise their analysis and adopt a more accepted threshold for seasonality assessment based on %RSV positive cases as opposed to the number of RSV cases to support a potential interpretation of “seasonal oscillation” (see also Midgley et al. 2017 JID 216(3):345-355).

5. Table 1: The authors should revise this Table to provide both numerators and denominators to allow for readability and logical follow with the main text. This will also allow the reader to appropriately follow the statistical assessment employed of relative proportions. In addition, Influenza prevalence is noted in the main text, but not in the corresponding Table 1. The authors should to revise the Table to include all relevant data for the reader.

6. Line 225-232 and Table 2: The authors should rephrase their statement regarding “clinical features of patients affect by RSV” to better reflect clinical characteristics of patients with SARI, since clinical data are presented for the total 632 patients and the 327 patients with RSV (180 RSV-A and 147 RSV-B). The numbers and percentages in the main text reflect the total population (N=632) and not the population of patients with RSV disease (N=327). The authors should further revise this Table to provide both numerators and denominators to allow for readability and logical follow with the main text. This will also allow the reader to appropriately follow the statistical assessment employed of relative proportions and to distinguish between RSV and everything else. Finally, viral load data in Table 2 is out of place without a (%) and should be included in Table 3 where viral load values are presented.

Minor Comments for Author (Required)

7. Line 17 and 40. The authors are repetitive in their statements in the Background and Conclusion sections of their Abstract regarding “understanding seasonality, genetic features…may support antiviral and vaccine development. The authors should revise the abstract and clarify how the results of this study specifically support antiviral and vaccine development

8. Lines 21, 38, 81, 83, 88, 339-440. Is the Brazilian Influenza Surveillance Program part of WHO’s Global Respiratory Syncytial Virus Surveillance Pilot and/or the Global Influenza Surveillance and Response System (GISRS)? The authors should consider revising for clarity; in particular Lines 338-340 at the start of the Discussion section where both programs are discussed in the context of the objectives of the current study. Recommend that the authors be consistent throughout the manuscript in their reference to the Influenza Surveillance Program as to which this study is based on (ie. National, Brazilian, or just Influenza Surveillance Program are used throughout the manuscript; pick one version and capitalize all words).

9. Line 30 and Line 105: What were the remaining 44% of case caused by, all influenza?

10. Line 48: The authors should clarify in the text the source of the “Influenza and other respiratory virus epidemiological reports” as to whether these are from the Brazilian and/or National Influenza Surveillance Program.

11. Line 57: The authors should explain the rationale as to why the previously observed significant association between viral load and disease severity should be more carefully studied in the Introduction. The authors later state in the Discussion that the correlation between viral load and disease severity remains controversial (Line 423). The authors are advised to further emphasize that one of the strengths of their study in finding of a lack of correlation between viral load and disease severity is the use of standardized methods for measuring viral load (see Lines 432-442)

12. Line 60: The authors should revise this sentence to clarify that the context by which “the treatment is based” in referring to RSV since this is new paragraph

13. Line 72: The authors should supplement reference 15 with a reference that defines the multiple genotypes of RSV-B

14. Line 78: Reference 15 does not support the statement that understanding RSV genetic diversity will help designing antiviral drugs, diagnostic assays, and vaccines. The authors should revise.

15. Fig 1: The y-axis and X-axis should be labeled within the figure.

16. Line 126-127: Location of INCAPER should be provided.

17. Line 143: The authors should define in Supplemental Table 1 or elsewhere in the main text what RSV gene the primers and probes used to subtype RSV-A and RSV-B were directed against

18. Line 161: The authors should clarify what they mean by “partial amplification” and by RSV positive samples with Ct values between 30-40 were not subjected or attempted for sequencing.

19. Line 179-180: The authors should provide a reference to the source of their reference sequences

20. Line 37, 74, 194, 294, 334, 421, 444, 466: The authors should correct their documentation of the RSV B genotype from BA to BA1 per the accession number provided and documented

6. PLOS authors have the option to publish the peer review history of their article (what does this mean?). If published, this will include your full peer review and any attached files.

Reviewer #1: No

Reviewer #2: No

---

## [Author Response · Author response to Decision Letter 0]

25 Jan 2021

Editor comments:

Comment 1: Please ensure that your manuscript meets PLOS ONE's style requirements, including those for file naming. The PLOS ONE style templates can be found at:

Answer: After a careful review, we have modified the formatting of the headings and legends of the supplementary figures and tables to meet PLOS ONE's style requirements. We also corrected some tables that presented values highlighted in red, contrary to the rules of PLOS ONE. Finally, we increased the font size of the Materials and methods, Results and Discussion subheadings to 16 pt, according to rules.

Comment 2: We note that you included minors (age<18) in your study. Please provide additional details regarding minor’s consent. In the ethics statement in the Methods and online submission information, please ensure that you have specified whether you obtained consent from parents or guardians. If the need for consent was waived by the ethics committee, please include this information.”

Answer: We agree with this observation. The sentence “The need for parents or guardians’ consent was waived by the ethics committee.” was included in “Ethics Statement” section. Please, check the lines 207-208.

Reviewer #1 comments:

1. Title - Revise the title to reflect the key findings of the research.

Answer: To address this comment, we have changed the title to: “Seasonality, molecular epidemiology and virulence of Respiratory Syncytial Virus (RSV): a perspective into the Brazilian Influenza Surveillance Program”. However, considering that this study addressed several aspects of RSV, the inclusion of key findings would make the title too large and we have opted to make it shorter and easier to read. 

2. Introduction - Revise the introduction and make it shorter and its current form it distracts the reader.

Answer: We have changed the introduction and eliminated sentences that, although interesting, would not affect the understanding of the objectives. However, Introduction size reduction was small. It turns out that this study has addressed multiple aspects of RSV (e.g.: prevalence, association between severity and subtypes, viral load, seasonality and association with climatic factors and phylogenetic aspects), as we understand that these are important aspects to discuss.

3. Methodology - Organize the Methodology - Study population, experimental methods - RT-qPCR, sequencing etc. in a brief form that gives a good understanding of the sequence of events with relevant methodology.

Answer: We have changed the way methods are presented, in order to maintain the same pattern presented at the results.

Thus, methods are now as follows:

1. Population sampling, study period and location

2. RSV and Influenza detection and subtyping

3. Clinical and epidemiological data collection

4. Viral load quantification

5. Climate data collection

6. Partial amplification and sequencing of glycoprotein gene

7. RSV genotyping and gene G phylogenetic reconstruction

8. Statistical treatment.

9. Data availability

10. Ethics Statement

4. Results - They need to be organized in the order of appearance as in the Methodology. You do not have to replicate all the information given in the Tables in the texts. Redundancy also distracts the reader and I found it difficult to organize the results to understand the authors' way of cohesion.

Answer: We think this comment will make the manuscript easier to read. As recommended, we have reorganized the objectives in the same pattern presented in the results, and eliminated redundant information.

5. Discussion - Again follow the order of your results in Discussion.

Answer: We have organized the discussion in the same order presented in Methods and Results and created subsections in the discussion, in order to improve reading.

6. Overall - You must revise this manuscript shortening certain sections and organizing the manuscript from I to D. Otherwise the results produced cannot be understood by the authors.

Answer: To address this recommendation, we have reduced the text as much as possible, without interfering with data presentation quality and consistency. We have reduced redundancies in the results (data being presented in the text and table) and removed some excerpts throughout the manuscript that we consider less relevant.

7. Language must be clear, correct, and unambiguous. At its current form it is difficult to follow the authors. Please also look into typographical or grammatical errors when your revise the manuscript. You may ask a native speaker to read the manuscript after fixing all the issues indicated.

Answer: we have asked a native speaker to thoroughly review the manuscript.

8. Please follow the PLOS ONE formatting guidelines well before you submit after revision.

Answer: We have done so.

Reviewer #2 comments:

1. The authors present generalized conclusions that are not specific to the Brazilian aspect or timeframe on which this study is based. Key results to support the identified objectives are not highlighted in the abstract or the conclusion (e.g. the influence of climate factors on RSV seasonality and the role of genetic diversity of RSV on disease severity). Clinical severity scores referenced in the abstract and results to support interpretations of the role of viral load and genetic diversity of RSV on disease severity, should be presented in the main tables/figures of the manuscript as opposed to supplemental. The authors should revise these areas and sharpen the focus of their Discussion through reduction to improve readability and presentation of key messages of RSV surveillance in Brazil between 2016-2018 relative to previous observations in Brazil or other parts of the world during the similar timeframe.

Answer: We appreciate the observations. However, some results of this study are not related to a specific location or timeframe. The correlation analyzes between viral load, genetic differences and severity are examples. These results possibly transcends the time and place of the study and, therefore, are not specific to the Brazilian aspect or timeframe.

As recommended, we have included the key results in both abstract and conclusion. We also transformed supplementary table 4 into Table 3. Previous table 3, which presented data on viral load, is now part of Table 4.

Given the different approaches taken in the study, we chose to divide the discussion into topics, in the hope of improving the quality of reading and regarding the discussion length, we removed some less important passages in order to improve the readability and presentation of key messages of the study.

2. The current title (and abstract) fail to address the presented timeframe of RSV surveillance or what aspects of “landscape” or “perspectives” the authors are referring to relative to their objectives and results. The authors should consider revision.

Answer: as changing the Title was also a recommendation of Reviewer 1, and to clarify which aspects of the “landscape” the study focused on, we have changed the title to: “Seasonality, molecular epidemiology and virulence of Respiratory Syncytial Virus (RSV): a perspective into the Brazilian Influenza Surveillance Program”. We hope this new title is suitable for both Reviewers.

3. Figure 2 and Figure 3 are out of focus and uninterruptable for review. The authors should revise.

Answer: Figures 2 and 3 were redone to improve quality and readability. We decided to change the size and save the file as .EPS.

4. Line 90-92, 130-133, and 270-276. The authors statement of “seasonal oscillation” (Line 90-92) is not supported by their main observations (Line 270-276) from Fig. 1, and in turn, their statement regarding “creating difficulties for determination of the most appropriate period to start prophylaxis” is not substantiated and is in contrast to their later statement of “recommends the administration of palivizumab from February to July” (Line 359). The authors state (Line 130-133) that “seasonality onset and end were defined as the first and last of 2 consecutive weeks, respectively, when the number of RSV cases exceeded 10% of the number detected during the RSV peak week” and reference Obando-Pacheco et al 2018 [21]. However, Obando-Pacheco et al 2018 states that “the onset of RSV season was defined as the first 2 consecutive weeks when >10% of the total tested samples for respiratory pathogens were positive for RSV. The end of the RSV season was defined similarly as when the proportion of positive RSV tests fell below 10% for 2 consecutive weeks.”. Given the impact of molecular testing on determining RSV seasonality, the authors should revise their analysis and adopt a more accepted threshold for seasonality assessment based on %RSV positive cases as opposed to the number of RSV cases to support a potential interpretation of “seasonal oscillation” (see also Midgley et al. 2017 JID 216(3):345-355).

Answer: We agree with the Reviewer, and, in fact, there was a misinterpretation of season beginning and end definition by Obando-Pacheco et al. (2018). Therefore, we reviewed the data and corrected the analysis. However, there were no changes in season onset in any year, but there were small changes in season end, as described below:

1. End in 2016: from EW 33 to EW 32.

2. End in 2017: from EW 30 to EW 31.

3. End in 2018: from EW 26 to EW 27.

Although the reviewer understood that the data do not support the claim that there was a fluctuation in season period during the study, we would like to point out that in 2016 and 2017 the RSV seasonal period started at epidemiological week (EW) 12 and ended at EW 32 and 31, respectively. In contrast, in 2018 the season was anticipated to EW 3, which is 9 weeks before the start in previous years. Season end was also anticipated to EW 27. There are the reasons why we understand there was an oscillation in the seasonal period during our study. As Palivizumab is administered in five consecutive monthly doses and considering that the first dose should be administered one month before season start, this oscillation may have an impact in the administration of prophylactic drugs.

In order to make this point clearer, we have restructured the discussion paragraphs.

5. Table 1: The authors should revise this Table to provide both numerators and denominators to allow for readability and logical follow with the main text. This will also allow the reader to appropriately follow the statistical assessment employed of relative proportions. In addition, Influenza prevalence is noted in the main text, but not in the corresponding Table 1. The authors should to revise the Table to include all relevant data for the reader.

Answer: we have revised Table 1, making the requested changes.

6. Line 225-232 and Table 2: The authors should rephrase their statement regarding “clinical features of patients affect by RSV” to better reflect clinical characteristics of patients with SARI, since clinical data are presented for the total 632 patients and the 327 patients with RSV (180 RSV-A and 147 RSV-B). The numbers and percentages in the main text reflect the total population (N=632) and not the population of patients with RSV disease (N=327). The authors should further revise this Table to provide both numerators and denominators to allow for readability and logical follow with the main text. This will also allow the reader to appropriately follow the statistical assessment employed of relative proportions and to distinguish between RSV and everything else. Finally, viral load data in Table 2 is out of place without a (%) and should be included in Table 3 where viral load values are presented.

Answer: Table 2 and the associated text present clinical data only of RSV infected patients. This table had an error in the "Sample number" field, which contained the total number of samples studied (632), however, the analyzes were performed only with the 352 RSV positive children. The error has been corrected. We took the opportunity to correct some fields containing three decimal places, standardizing the values to two decimal places. We have also relocated table 2 viral load data to table 3, as suggested. Finally, we included denominators to facilitate data interpretation.

Minor Comments for Author (Required)

7. Line 17 and 40. The authors are repetitive in their statements in the Background and Conclusion sections of their Abstract regarding “understanding seasonality, genetic features…may support antiviral and vaccine development. The authors should revise the abstract and clarify how the results of this study specifically support antiviral and vaccine development.

Answer: We have eliminated the redundant part from the "Background" and briefly discussed how seasonal period, virulence and genetic diversity can assist in the development and application of vaccines and antiviral drugs.

8. Lines 21, 38, 81, 83, 88, 339-440. Is the Brazilian Influenza Surveillance Program part of WHO’s Global Respiratory Syncytial Virus Surveillance Pilot and/or the Global Influenza Surveillance and Response System (GISRS)? The authors should consider revising for clarity; in particular Lines 338-340 at the start of the Discussion section where both programs are discussed in the context of the objectives of the current study. Recommend that the authors be consistent throughout the manuscript in their reference to the Influenza Surveillance Program as to which this study is based on (ie. National, Brazilian, or just Influenza Surveillance Program are used throughout the manuscript; pick one version and capitalize all words).

Answer: The Brazilian Influenza Surveillance Program is part of WHO’s Global Respiratory Syncytial Virus Surveillance. We chose to use the term “Brazilian Influenza Surveillance Program” with capital words, as suggested.

9. Line 30 and Line 105: What were the remaining 44% of case caused by, all influenza?

Answer: In this study, only RSV and Influenza were tested. In 56% of the cases RSV was detected, the influenza virus was found in 7% of the samples and the remaining 37% cases were undetermined.

10. Line 48: The authors should clarify in the text the source of the “Influenza and other respiratory virus epidemiological reports” as to whether these are from the Brazilian and/or National Influenza Surveillance Program.

Answer: We have followed previous Reviewers’ recommendations to shorten the introduction, and because of that we have removed this text, as explained to the other reviewer.

11. Line 57: The authors should explain the rationale as to why the previously observed significant association between viral load and disease severity should be more carefully studied in the Introduction. The authors later state in the Discussion that the correlation between viral load and disease severity remains controversial (Line 423). The authors are advised to further emphasize that one of the strengths of their study in finding of a lack of correlation between viral load and disease severity is the use of standardized methods for measuring viral load (see Lines 432-442)

Answer: To address that, we have rewritten the Introduction as follows:

“Some studies have evaluated the association between viral load and disease severity, with significant associations [6,7]. However, most of these studies did not use standardized methods of viral load measurement, therefore, this relationship must be more carefully evaluated.”

12. Line 60: The authors should revise this sentence to clarify that the context by which “the treatment is based” in referring to RSV since this is new paragraph.

Answer: We have rephrased the sentence to: “RSV treatment is based only […]”.

13. Line 72: The authors should supplement reference 15 with a reference that defines the multiple genotypes of RSV-B.

Answer: We have added the study by Trento et al. (2006) which was already mentioned in reference #17 (now reference #14, since due to the removal of some sections to reduce the text, the corresponding references were also removed).

14. Line 78: Reference 15 does not support the statement that understanding RSV genetic diversity will help designing antiviral drugs, diagnostic assays, and vaccines. The authors should revise.

Answer: It is possible to find in reference 15 (now reordered to reference 13) two excerpts that support this statement: “RSV diversity is an important factor that allows for reinfections to occur throughout life and also has implications for design of diagnostic assays, antiviral therapies, and preventive strategies (passive immunization and vaccines)”. (in the introduction). 

“Genotype classification and assignment is of importance in order to understand the evolution, epidemiology, and clinical presentation of this virus, and has implications regarding the development of vaccines and other preventive interventions.” (in the discussion). 

15. Fig 1: The y-axis and X-axis should be labeled within the figure.

Answer: Figure 1 has been edited, including caption for the two Y axes and the X axis. Caption is displayed in a text box.

16. Line 126-127: Location of INCAPER should be provided.

Answer: We have included the city, state and country of INCAPER. Please, check the line 145.

17. Line 143: The authors should define in Supplemental Table 1 or elsewhere in the main text what RSV gene the primers and probes used to subtype RSV-A and RSV-B were directed against.

Answer: In the methodology, we include the requested information as follows: “RSV positive samples (i.e. those with cycle threshold [CT] ≤ 40) were subtyped using specific primers and probes to N gene of RSV-A and RSV-B.” Please, check the line 105.

18. Line 161: The authors should clarify what they mean by “partial amplification” and by RSV positive samples with Ct values between 30-40 were not subjected or attempted for sequencing.

Answer: Partial amplification in this case refers to the fact only part of the gene was amplified. We have included the approximate sequenced G gene fragment size, as follows: 

“The partial gene G amplification (about 730 bp) was performed at LVRS/IOC/FIOCRUZ”

We have also included the following sentence in bold: “a) cycle threshold (ct) value less than 30, due to the difficulty in sequencing samples with higher ct than this;”

19. Line 179-180: The authors should provide a reference to the source of their reference sequences.

Answer: The requested data is already available in supplementary tables 2 and 3. All reference sequences were taken from NCBI Genbank. These supplementary tables contain access numbers, genotypes and collection locations of each sequence.

20. Line 37, 74, 194, 294, 334, 421, 444, 466: The authors should correct their documentation of the RSV B genotype from BA to BA1 per the accession number provided and documented.

Answer: The classification into the BA cluster is controversial. We prefer classify as BA. More studies are needed to standardize the RSV nomenclature of genotypes into BA and ON1.

---

## [Decision Letter · Decision Letter 1]

15 Mar 2021

PONE-D-20-30854R1

Seasonality, molecular epidemiology and virulence of Respiratory Syncytial Virus (RSV): a perspective into the Brazilian Influenza Surveillance Program

PLOS ONE

Dear Dr. Vianna,

Thank you for submitting your manuscript to PLOS ONE. After careful consideration, we feel that it has merit but does not fully meet PLOS ONE’s publication criteria as it currently stands. Therefore, we invite you to submit a revised version of the manuscript that addresses the points raised during the review process.

The reviewers agreed that the revised manuscript showed significant improvement, however, one reviewer still have concern regarding clinical data which needs to be address carefully.  In addition, I have found quite a few typos and error within the manuscript that need to be corrected (see attached PDF file from editor)

We look forward to receiving your revised manuscript.

Kind regards,

Baochuan Lin, Ph.D.

Academic Editor

PLOS ONE

Journal Requirements:

Reviewers' comments:

Reviewer's Responses to Questions

**Comments to the Author**

1. If the authors have adequately addressed your comments raised in a previous round of review and you feel that this manuscript is now acceptable for publication, you may indicate that here to bypass the “Comments to the Author” section, enter your conflict of interest statement in the “Confidential to Editor” section, and submit your "Accept" recommendation.

Reviewer #3: All comments have been addressed

Reviewer #4: All comments have been addressed

2. Is the manuscript technically sound, and do the data support the conclusions?

Reviewer #3: Yes

Reviewer #4: Yes

3. Has the statistical analysis been performed appropriately and rigorously? 

Reviewer #3: No

Reviewer #4: Yes

4. Have the authors made all data underlying the findings in their manuscript fully available?

Reviewer #3: No

Reviewer #4: Yes

5. Is the manuscript presented in an intelligible fashion and written in standard English?

Reviewer #3: Yes

Reviewer #4: Yes

6. Review Comments to the Author

Reviewer #3: The clinical dataset does not appear to have been uploaded in the way the genetic data has. Some of the statistical choices are suboptimal but overall this is decent if unglamorous science that fits the PLoS model

Reviewer #4: No further comments to strengthen the paper. Thank you for allowing me to review your paper and thanks for making the changes.

7. PLOS authors have the option to publish the peer review history of their article (what does this mean?). If published, this will include your full peer review and any attached files.

Reviewer #3: **Yes: **Paul Walsh

Reviewer #4: No

---

## [Author Response · Author response to Decision Letter 1]

24 Mar 2021

RESPONSE TO REVIEWERS

Editor’s comments:

The reviewers agreed that the revised manuscript showed significant improvement, however, one reviewer still have concern regarding clinical data, which needs to be address carefully. In addition, I have found quite a few typos and error within the manuscript that need to be corrected (see attached PDF file from editor).

Answer: Dear Editor, thank you for considering our manuscript. We thoroughly checked comments in the PDF file and made all suggested corrections

Comment 1: About the sentence: "RSV viral load was determined by RT-qPCR using a protocol adapted from Álvarez-Argüelles et al. [20], including a synthetic β-globin dsDNA as a template.”, it was asked if the β-globin dsDNA was used as an internal PCR control.

Answer: β-globin dsDNA was used for sample cell quantification; using the same standard curve construction method applied for measuring RSV copies, we were able to calculate the number of cells in the sample. Applied Biosystems 7500 Real-Time PCR Software is able to use various known concentrations of β-globin (comparing these curves with the sample amplification curve) to estimate number of β-globin copies in the sample. This number can later be used to calculate the number of cells in a sample. In time, the marker used as an internal control was the human RNAse P gene.

Comment 2: About the sentence: " Male gender was 220 slightly more affected by RSV (n=182; 52%) […]” it was asked whether the finding was statistically significant. 

Answer: Negative. The difference found in the proportion between male and female patients affected by RSV presented no statistical support. However, as other authors commonly present these data, we decided to show it as well. In addition, we compared viral load between genders.

Comment 3: About the sentence: “S4 Table shows the difference in severity by ethnicity.”, it was asked for more description.

Answer: To address the request of reviewer #3 (Comment n°9), we decided to eliminate discussions about race from the manuscript, and, therefore, the S4 table was removed.

Comment 4: Please review your reference list to ensure that it is complete and correct. If you have cited papers that have been retracted, please include the rationale for doing so in the manuscript text, or remove these references and replace them with relevant current references. Any changes to the reference list should be mentioned in the rebuttal letter that accompanies your revised manuscript. If you need to cite a retracted article, indicate the article’s retracted status in the References list and also include a citation and full reference for the retraction notice.

Answer: We use the Zotero software as an indexer of bibliographic references. In this way, all the works cited are listed automatically. Even so, we revised the manuscript to ensure that all and only the papers cited are referenced. 

Once the discussion in the context of race was removed from the manuscript, the references used were also eliminated, namely:

• Reference 31 (Cardena et al., 2005);

• Reference 32 (Brasil. Síntese de Indicadores Sociais: uma análise das condições de vida da população brasileira);

• Reference 33 (Nair et al., 2010)

We also added a reference (de-Paris et al., 2014) in order to include data on length of hospital stay in the discussion. This measure aimed to meet the recommendation of reviewer #3 to emphasize the data on duration of hospitalization found in our study.

Two other references (Thomas, 2013 [ref. 30] and Wonderlich et al, 2017 [ref. 31]) were added in order to support the discussion suggested by the reviewer in his comment No. 17.

We did not use papers that have been retracted.

Reviewer #3 comments:

Topic n°1: Sample selection

Comment n°1: How were the municipalities selected? Was sampling representative? And if so was a weighting scheme used that would allow improved generalization of the results? Given the international audience of PLoS, I would favor including a map with distance marker, topography and major roads be provided perhaps this could be done as a link to a google map page would be helpful and avoid the impression of being a gimmick to Brazilian readers while helping international ones.

Answer: Most of our study was carried out in the Molecular Biology II sector of the Central Laboratory of Espirito Santo (LACEN/ES). This government agency is part of the national laboratory surveillance network for Influenza and other respiratory viruses and, for this reason, has the duty to investigate all cases that fit the SARI concept adopted in Brazil. Thus, any sample collected in this context, even if it comes from private institutions, must be sent to LACEN for diagnosis of Influenza, Respiratory Syncytial Virus and, most recently, of SARS CoV 2. Therefore, no sampling or weighting scheme was used, for the simple fact that virtually all samples of children up to 3 years old with SARI in Espirito Santo were present in our study. Consequently, the sampled municipalities were all those with suspected cases (61 municipalities). Of these, 46 presented at least one child with a confirmed case of RSV infection.

We have built a map of the state of Espirito Santo (S1 Fig.), with the main highways (federal highways), distance marker and topography, as requested. We plotted the number of confirmed RSV cases in each municipality throughout the study period. We are grateful for the idea, which will help the international community better understand the geographic space of the study.

Comment n°2: Median length of stay was high at 8 days suggesting this was a sicker than usual group of infants with RSV. The potential for this later result should be signaled in the sampling scheme.

Answer: To address this comment, we have included the following text:

"Although the median length of hospitalization found here is similar to some other studies [28,29], notably most studies report a shorter duration [1,4,26]. One hypothesis that could explain this finding is the fact that all children included in our study were diagnosed with SARI, which makes our study group a cohort with severe RSV infection. Another hypothesis is linked to the possibility that most of the children in the study had an infection in the lower respiratory tract. Aerosol transmission increases the chances of inhaling viral particles in the lower airways, while larger droplets are retained in the upper airways [30]. Naturally, aerosol infections tend to trigger a more severe course of infection [31].”

We would like to point that there was no sampling scheme, as all the samples in the Espirito Santo State are sent to us for analysis.

Topic n°2: Clinical Outcomes and study definitions

Comment n°3: Dichotomizing 02 saturation, especially at 95% which is not a clinically used threshold seemed odd. 90%, 92% or even 94% would be consistent with variously described clinical thresholds.

Answer: This is a retrospective study in which all clinical data were taken directly from the SARI epidemiological records, used by the Brazilian health system. In these forms, there is a field with the following question: “Oxygen saturation ≤ 95%?”. There is no field in the records for filling with the measured saturation per se, but only if it was above, equal or below 95%. It is, therefore, qualitative data. The reviewer very well pointed out that the dichotomization of O2 saturation is not commonly clinically used; however, unfortunately, this was the only possible way of analysis, given the data available. However, it is necessary to say that saturation ≤95% is used as an element in the definition of SARI in Brazil and, although it is not the most common form found in studies, it is used as an indicator of severity in respiratory diseases.

Comment n°4: CSS score seems very crude. Was it derived and validated before this study? If not, it is simply reframing the data already gathered and would be better analyzed as discrete variables rather than as a score. It would be better to analyze the CSS itself as an ordinal variable. This would also increase the statistical power of the analysis.

Answer: The CSS used in our study was adapted from Martinello and colleagues, 2002 (see the comparison in the table below). The adaptations were made due to subtle differences in the way data was obtained. For example, the way of measuring oxygen saturation in our study, as explained in response to comment No. 3, was obtained in a binary way (yes or no) in relation to a saturation lower than 95%. In contrast, Martinello et al., 2002 used the threshold of 87%. CSS was a way that we found to combine the data most associated with disease severity in a single parameter. It is the use of the basic principle of "reduction of dimensionality" of the problem. In our view, CSS was treated as an ordinal categorical variable, and not as a discrete variable. In fact, CSS cannot be used as a discrete value since "score # 2" is not twice as severe as score # 1, for example.

 Martinello et al., 2002 This study

Assessed factors Number of points

Mechanically ventilator support 2 2

Use of supplemental oxygen 1 1

Hospital admission 1 - *

Hospitalization ≥ 5 days 1 1

Oxygen saturation ≤ 87% 1 1

ICU admission - 1

Range 0-6 points 0-5 points**

* Since virtually all patients in our study were diagnosed with SARI, they were all hospitalized. In this way, everyone would receive 1 point on the scale used by Martinello et al., 2002. As only the most serious cases required admission to ICU, we modified the score for this situation.

** In our study, a patient who was initially treated with oxygen supplementation, however, due to the worsening of the case, required mechanical ventilation received 2 points and not 3 for the fact that he was, technically, in both categories.

Comment n°5: Dyspnea (p=0.148) and respiratory distress (p=0.002) seem difficult to distinguish to me. Do the authors mean tachypnea? If so, what rate/age thresholds were used? Invasive versus non-invasive 02 therapies.

Answer: The SARI notification form used by the Brazilian health system includes both parameters: dyspnea and respiratory distress, that should be informed by the doctor or nurse during the patient's care, and yes, these are synonyms. Dyspnea is a more technical term that, according to the American Thoracic Society (ATS), is defined as a subjective experience of respiratory distress. But as people can generally only describe what they can understand, a more lay term is also present in the notification. Our group discussed the possibility of eliminating one term, but opted to keep both terms in the analysis, since this is how data was originally collected and analyzed.

Topic n°3: Viral load and clinical outcomes

Comment n°6: “According to age, median viral load was higher in children with 4 to 6 months old (63.0 261 cop/cell, p=0.007). Regarding patient clinical conditions, we found lower viral load I 262 patients with fever (26.15 cop/cell) than those without it (111.29 cop/cell; p=0.00)”. A key point is this: were those children who had no fever treated with antipyretics? If so then maybe this is an effect of the antipyretic which has been shown to increase viral shedding in animal models of RSV and duration of shedding in rhinovirus. If they were not the implication may be that fever was reflecting an antibody response to higher viral loads. (Also, write this as p<0.01.)

Answer: All clinical information was obtained from SARI notification forms. In these forms, in most cases, the drugs used in the treatment are not listed. That is why it is impossible to retrieve information about which patients used antipyretics or not. Thus, although it was a very interesting recommendation by the reviewer, unfortunately it is not possible to carry out this type of analysis.

Comment n°7: (Table 4 and line 265) “Although lacking statistical support (p=0.089), a noteworthy observation is the tendency of lower viral load in patients with elevated CSS.” The viral load analysis was performed regardless of time between symptom onset and date of collection, which, in theory, could alter the interpretation. However later it appears that they did a sensitivity analysis implying Day 7 measures had essentially the same results as other time points? It was unclear to me exactly what they mean here. 

Answer: The CSS was calculated based on parameters generally associated with the severity of the infection, such as the need for intensive therapies, the use of mechanical ventilation, hospitalization, etc. When comparing the median viral load between the different clinical severity scores, we noticed a tendency for the lowest scores (less severe disease) to have a higher viral load. This observation could indicate an inversely proportional relationship between viral titers and the severity of the disease. Although it seems to be an unexpected relationship, other authors found very similar results (Haynes et al., 2013; Piedra et al., 2017). Knowing that the tendency of viral load is to decrease as the disease progresses, a possible common doubt to the reader would be whether these findings could not be associated with a bias caused by different sample collection dates. Therefore, we performed a serial analysis in which the results were very similar to the result using the sampling indistinctly from the date of collection. However, in our study there was no statistical significance for the relationship found, which limits us to go further in the discussion.

Comment n°8: The authors also found fever and viral load to be inversely related. This is important because it is counterintuitive at first blush but makes complete sense when interpreted in the context of RSV being capable of infecting via aerosols reaching the alveoli as well as droplets seeding the upper airways. If lung ultrasound data were available that would be an interesting way to address this apparent paradox. (Probably requires another paper though.)

Answer: Once again, we would like to thank the reviewer's brilliant insights. However, unfortunately, we do not have lung ultrasound data on patients. This analysis, nevertheless, can be the focus of a new prospective study for more robust investigations in this and other aspects.

Topic n°4: Race

Comment n°9: The authors’ use of race is baffling to me. Rather than using race which is suspect at the best of times, and per their discussion, especially in Brazil, could the authors use SES? If race is important, and their own discussion suggests otherwise, some classification other than color is needed. If it can’t be explained it should be dropped.

Answer: In fact, the inclusion of this discussion in the study raised doubts even among the authors. The idea would be to bring more data about a topic that has already been addressed in other studies (doi:10.1542/peds.2004-0059) and not to cause discomfort to readers. This comment was important to reinforce that the comparison between races, although in the best intention of evaluating possible differences between ancestry and response to infections, is inappropriate. In addition, as the reviewer rightly stated, our data do not suggest that there is a direct relationship between the severity of the infection and ethnicities. So, we appreciate the feedback and decided to remove this topic from the study.

Topic n°5: Climate data

Comment n°10: Line 368 on Precipitation rate and relative humidity percentage have not been shown to influence the distribution of RSV cases by Spearman's correlation test (p = 0.55 and 0.11, respectively). The mean temperature, however, showed a minor and inverse correlation with RSV infections (-0.16; p = 0.05). Please use 'did not' rather 'have not been shown' if these are your findings rather than someone else’s. If they are from someone else, please include the reference.

Answer: Those are our findings and, therefore, we made the correction as advised.

Topic n°6: Table 1

Comment n°11: Unclear why there are P-values in Table 1 unless the authors are testing a hypothesis that there were important differences between seasons. If they are, then they should say this and incorporate season as a variable in the final model

Answer: The reviewer's comment is pertinent. Yes, the p-value in table 1 resulted from a comparison of the scenarios between the seasons, which shows that some of the difference observed is not accidental. However, due to the length of the manuscript and following the suggestion of the other reviewers to decrease the size of the discussion and sharpen the focus in some areas, we chose not to discuss these results. Taking into account that data without discussion adds little to scientific knowledge, we chose to remove this statistical data from the table, which will make it cleaner and easier to interpret. We are grateful for the observation.

Comment n°12: (Comment about Table 1) The authors report (%) implying that percentages are in parentheses however it appears that it is the proportions they are reporting in parentheses.

Answer: We would like to thank the reviewer for the observation. Certainly, the header of the tables indicated that the relationship would be presented in percentage, but the data were presented in proportion. We changed the presentation to the percentage format (%) as shown in the table.

Topic n°7: Table 2

Comment n°13: This is important. I would have placed death with the outcomes/clinical profile. In demographics it is sufficient to provide the number for one gender only. Indicating the use of a specific test as a Table foot note is needed only if not already in the methods or is less commonly used.

Answer: As advised, we moved the RSV death data to the "Clinical profile" category, kept the gender data for a single gender and removed the table foot note containing the statistical tests performed, since they were frequently used methods.

Topic n°7: Table 3

Comment n°14: Table 3 appears to analyze CSS as categorical rather than ordinal data. Please be consistent with format for decimals; either, or. but do not use them interchangeably. I expect PLoS favors the US/UK format of ‘.’. It would be better to present the components of the CSS individually and the CSS itself as an ordinal variable. This would also increase the statistical power of the analysis.

Answer: As reported in comment n°4, CSS was analyzed as an ordinal variable. In addition, the components used in the construction of the CSS were treated individually in the comparison between the subtypes RSV-A and RSV-B (Table 2). Finally, we inform that we have corrected the field in which there was a "," instead of a “.” in table 3. All other numbers followed the US / UK format.

Topic n°8: Table 4

Comment n°15: Same comments as for prior tables apply. Days in intensive care should be compared using a survival analysis technique. If a p value appears as ‘= 0.000’ in computer software, please report it as p <0.001.

Answer: As requested, we performed the survival curve of patients who required admission to the ICU and, for this, we used the non-parametric Kaplan-Meier test. However, considering that of the patients with data on ICU entry and exit, only 5 died and 134 had “recovery” as an outcome, we used the cure outcome and not death as usual as "failure". That is, the test revealed that the average time until the recovery of children admitted to ICU was 8 days, from the first day of intensive care. From this, we used the Cox regression (or proportional hazards regression) in order to assess whether the viral load would have a statistically significant effect on the duration of ICU stay. Since the p-value was greater than 0.05, we can infer that there is no such effect. We emphasize that we use Schoenfeld residuals and that the proportional hazards assumption were met. We also modified presentation of the p-values as suggested by the reviewer.

Topic n°8: Figure 1

Comment n°16: Please add a second X -axis below the current one to indicate season or month. I have not been able to download the supplemental materials – I just get a pdf of the submission.

Answer: Below the X-axis we inserted three dividing lines that represent each of the years (seasons) of the study.

Topic n°9: Discussion

Comment n°17: An explanation not raised by the authors is that maybe these children had predominantly lower respiratory tract disease. This could arise where the bulk of the viral transmission occurred via the airborne rather than the large droplet route and that some of the infection started in the alveoli rather than the larger airways. This would lead to a more rapid onset and more severe disease. See above comments for the other areas of concern. This is slightly more nuanced than the previous reviewers who argue the data shows no correlation between viral load and disease. I think the nuance is warranted.

Answer: Initially, we had doubts about which point in the discussion of the results the reviewer wanted to address that question. However, we believe that it fits properly in the discussion regarding the prolonged hospital stay observed in our study. Thus, we included the following text (see answer to comment n°2): 

“Although the median length of hospitalization found here is similar to some other studies [28,29], notably most studies report a shorter duration [1,4,26]. One hypothesis that could explain this finding is the fact that all children included in our study were diagnosed with SARI, which makes our study group a cohort with severe RSV infection. Another hypothesis is linked to the possibility that most of the children in the study had an infection in the lower respiratory tract. Aerosol transmission increases the chances of inhaling viral particles in the lower airways, while larger droplets are retained in the upper airways [30]. Naturally, aerosol infections tend to trigger a more severe course of infection [31].”

Topic n°9: Clinical dataset

Comment n°18: The clinical dataset does not appear to have been uploaded in the way the genetic data has. Some of the statistical choices are suboptimal but overall this is decent if unglamorous science that fits the PLoS model.

Answer: In order to meet the reviewer's requirements, we decided to upload the table with general data (including clinical data) as a supplementary table (S4 Table) in the materials and methods section. In this way, clinical data are also accessible as suggested by the reviewer. Regarding the statistical analysis, we performed the survival analysis recommended in comment n°15.

---

## [Decision Letter · Decision Letter 2]

13 Apr 2021

PONE-D-20-30854R2

Seasonality, molecular epidemiology and virulence of Respiratory Syncytial Virus (RSV): a perspective into the Brazilian Influenza Surveillance Program

PLOS ONE

Dear Dr. Vianna,

Thank you for submitting your manuscript to PLOS ONE. After careful consideration, we feel that it has merit but does not fully meet PLOS ONE’s publication criteria as it currently stands. Therefore, we invite you to submit a revised version of the manuscript that addresses the points raised during the review process.

While the revised manuscript is scientifically sound, I have a few comments that need to be addressed. 1. Line 61-62, only one study is cited, so please change "..., other studies..." to "..., other study..." 2. S1 Table, please include PCR conditions.  3. Line 171, there is no need to cite reference 22 , suggest delete.  Please change "...Sequencher 5.1 [22]." to "...Sequencher 5.1 (Gene Codes Corporation, Ann Arbor, MI, USA).  4. Line 224, please delete the statement "Male gender was slightly more affected by RSV", since it is not statistically significant and not supported by the data from 2016 (Table 1). 5. Line 232 - 235, are the number of 341, 307, 336, 342 and 252 out of the total 632 patients? I am trying to clarify whether the authors mean that out of 632 patients, 341 experienced cough and 318 (93%) are RSV+ etc.? 6. Line 272 - 274, not sure what the authors wish to convey, this sentence needs rephrasing for clarity. 7. Figure legends for figures 2&3 need correction since no highlighted in bold in the figures. 8. Line 516 - 518, delete this sentence or move to the very beginning of the conclusion.

Additionally the quality of language still needs improvement.  We suggest you thoroughly copyedit your manuscript for language usage, spelling, and grammar. If you do not know anyone who can help you do this, you may wish to consider employing a professional scientific editing service.

We look forward to receiving your revised manuscript.

Kind regards,

Baochuan Lin, Ph.D.

Academic Editor

PLOS ONE

Journal Requirements:

Reviewers' comments:

Reviewer's Responses to Questions

**Comments to the Author**

1. If the authors have adequately addressed your comments raised in a previous round of review and you feel that this manuscript is now acceptable for publication, you may indicate that here to bypass the “Comments to the Author” section, enter your conflict of interest statement in the “Confidential to Editor” section, and submit your "Accept" recommendation.

Reviewer #3: All comments have been addressed

2. Is the manuscript technically sound, and do the data support the conclusions?

Reviewer #3: Yes

3. Has the statistical analysis been performed appropriately and rigorously? 

Reviewer #3: Yes

4. Have the authors made all data underlying the findings in their manuscript fully available?

Reviewer #3: No

5. Is the manuscript presented in an intelligible fashion and written in standard English?

Reviewer #3: Yes

6. Review Comments to the Author

Reviewer #3: I can't edit the manuscript for typos but the authors have addressed my concerns to the extent possible in their available data, and what cannot be answered should not delay the paper further.

7. PLOS authors have the option to publish the peer review history of their article (what does this mean?). If published, this will include your full peer review and any attached files.

Reviewer #3: **Yes: **Paul Walsh

---

## [Author Response · Author response to Decision Letter 2]

21 Apr 2021

We thank the editor for the critical assessment of our manuscript. In the following we address the concerns point by point.

Editor’s comments:

While the revised manuscript is scientifically sound, I have a few comments that need to be addressed. 

Comment n°1: Line 61-62, only one study is cited, so please change "..., other studies..." to "..., other study..."

Reply: We have done so.

Comment n°2: S1 Table, please include PCR conditions. 

Reply: As suggested, we include thermocycling conditions for each set of primers and probes used.

Comment n°3: Line 171, there is no need to cite reference 22, suggest delete. Please change "...Sequencher 5.1 [22]." to "...Sequencher 5.1 (Gene Codes Corporation, Ann Arbor, MI, USA). 

Reply: We removed the reference and included the recommended excerpt.

Comment n°4: Line 224, please delete the statement "Male gender was slightly more affected by RSV", since it is not statistically significant and not supported by the data from 2016 (Table 1). 

Reply: We have done so.

Comment n°5: Line 232 - 235, are the number of 341, 307, 336, 342 and 252 out of the total 632 patients? I am trying to clarify whether the authors mean that out of 632 patients, 341 experienced cough and 318 (93%) are RSV+ etc.? 

Reply: We are grateful for the editor's observation, as this would possibly be a common question for readers. Therefore, we decided to include the following excerpt in the caption of Table 2:

“Although the study included 352 patients with RSV, it is possible to observe that the denominators in the clinical profile differ from this number. This occurred because not all clinical data were recorded for all patients.”

Comment n°6: Line 272 - 274, not sure what the authors wish to convey, this sentence needs rephrasing for clarity. 

Reply: We rephrased the sentence as follows:

“However, a segmented analysis (0-3; 4-7 and >7 days between symptom onset and sample collection) revealed very similar results. Furthermore, of the 156 samples used to measure viral titers, only 26 (16%) were collected 7 days after symptoms onset. Therefore, we prefer to maintain full sampling for viral load analysis.”

Comment n°7: Figure legends for figures 2&3 need correction since no highlighted in bold in the figures. 

Reply: Fixed.

Comment n°8: Line 516 - 518, delete this sentence or move to the very beginning of the conclusion.

Reply: The section was deleted.

Comment n°9: Additionally the quality of language still needs improvement. We suggest you thoroughly copyedit your manuscript for language usage, spelling, and grammar. If you do not know anyone who can help you do this, you may wish to consider employing a professional scientific editing service.

Reply: Manuscript editing was performed by other two researchers with proficiency in the English language and also by a professional reviewer with experience in scientific journals. We hope that this time the spelling, grammar, and language are in perfect harmony with the formal language.

---

## [Editor Report · Decision Letter 3]

26 Apr 2021

Seasonality, molecular epidemiology, and virulence of Respiratory Syncytial Virus (RSV): a perspective into the Brazilian Influenza Surveillance Program

PONE-D-20-30854R3

Dear Dr. Vianna,

We’re pleased to inform you that your manuscript has been judged scientifically suitable for publication and will be formally accepted for publication once it meets all outstanding technical requirements.

Kind regards,

Baochuan Lin, Ph.D.

Academic Editor

PLOS ONE
---

## [Editor Report · Acceptance letter]

4 May 2021

PONE-D-20-30854R3 

Seasonality, molecular epidemiology, and virulence of Respiratory Syncytial Virus (RSV): a perspective into the Brazilian Influenza Surveillance Program 

Dear Dr. Vianna:

I'm pleased to inform you that your manuscript has been deemed suitable for publication in PLOS ONE. Congratulations! Your manuscript is now with our production department. 

Kind regards, 

on behalf of

Dr. Baochuan Lin 

Academic Editor

PLOS ONE